# Mislabeled examples detection viewed as probing machine learning models: concepts, survey and extensive benchmark

**Thomas George**[*]                    *thomas.george@orange.com*
*Orange Innovation*

**Pierre Nodet**[*]                    *pierre.nodet@orange.com*
*Orange Innovation*

**Alexis Bondu**                    *alexis.bondu@orange.com*
*Orange Innovation*

**Vincent Lemaire**                    *vincent.lemaire@orange.com*
*Orange Innovation*

Reviewed on OpenReview: *https://openreview.net/forum?id=3YlOr7BHkx*

## Abstract

Mislabeled examples are ubiquitous in real-world machine learning datasets, advocating the development of techniques for automatic detection. We show that most mislabeled detection methods can be viewed as probing trained machine learning models using a few core principles. We formalize a modular framework that encompasses these methods, parameterized by only 4 building blocks, as well as a Python library that demonstrates that these principles can actually be implemented. The focus is on classifier-agnostic concepts, with an emphasis on adapting methods developed for deep learning models to non-deep classifiers for tabular data. We benchmark existing methods on (artificial) Completely At Random (NCAR) as well as (realistic) Not At Random (NNAR) labeling noise from a variety of tasks with imperfect labeling rules. This benchmark provides new insights as well as limitations of existing methods in this setup.

## 1 Introduction

In supervised machine learning, the performance of learned algorithms crucially depends on the quality of the dataset of examples used during training: how many examples do we have access to, are these examples representative of the actual distribution on the feature space, and were the training examples correctly labeled. We focus on the latter subject. Indeed, many actual use cases include some amount of labeling errors. For example, this is typically the case in tasks that involve human supervision since labeling large datasets requires a pool of annotators that possess a mix of expert knowledge (which is costly), and willingness to perform repetitive tasks (which is dull). This is also known to be the case for widely used benchmark datasets such as CIFAR10/100 or MNIST (Northcutt et al., 2021b). Therefore, cleansing datasets offers the promise of better performance, but at the cost of additional efforts. Since the early days of machine learning, it has been widely believed that this could be achieved through automated methods, eliminating the need for further human intervention. This has led to a number of methods for automatic detection of mislabeled examples using classical machine learning methods (Guan & Yuan, 2013). With the success of deep learning methods in applications ranging from image recognition to language models, new mislabeled detection methods have also been proposed that exploit its specific training dynamics.

---

[*]equal contribution

The aim of this paper is to offer a new perspective on existing mislabeled detection methods, as well as practical recommendations in actual use cases in the presence of labeling noise. Rather than learning a model that captures the structure of the labeling noise, our approach is to blindly evaluate existing methods on real-world datasets, with no prior knowledge. We survey mislabeled detection methods regardless of whether they were designed to work with deep learning models or other classical machine learning algorithms, and we highlight a few common principles. We also focus on tabular and text data, a type of data that is prevalent in the industry (e.g. in logs, in customer databases, etc) but that has recently received less attention than datasets that are more amenable to deep learning methods such as images, sound or language.

This paper is organized as follows: In **section 2**, we suggest a definition of the problem of detecting mislabeled examples. We discuss its limitations and the inherent ambiguity of what can be considered a mislabeled example in the statistical learning framework. We also describe existing strategies for dealing with mislabeled examples. In **section 3**, we highlight the concepts behind a majority of mislabeled detection methods found in the literature. We show that most methods can be described using a few core principles with 4 distinct components. We survey a large amount of existing mislabeled detection methods and show how they fit in this framework. The modularity of this framework can also be leveraged to design new methods by recombining existing components. We review a number of extensions to existing methods that address some specific issues encountered in real-world use cases. In **section 4**, we describe our contributed library that implements the framework of the previous section, showing that it is not just a theoretical view but that it can actually be implemented. Most existing methods in the literature can be readily implemented using this library. In **section 5**, we perform a large-scale experiment by varying detectors, handling strategies, and noise structure on a large number of actual datasets. This leads us to new insights regarding the behavior of mislabeled detection methods in different setups, as well as practical guidance aimed at future practitioners. **Section 6** is dedicated to other related works that were not included in previous sections, and we conclude in **section 7**.

**List of contributions**

- **Concepts** – We provide a fresh view on a majority of existing mislabeled detection methods by showing that they can be described in a modular framework that uses a few components. This allows us to provide a large survey the highlights the similarities and differences of these methods. We identify a set of probes that provide the base signal used to discriminate between genuine and mislabeled examples.

- **Survey** – We review a large number of existing detection methods, as well as other strategies to address specific issues. We survey 3 common strategies in weakly supervised learning.

- **Implementation** – We contribute a library that allows instantiating a large number of existing mislabeled detection methods, as well as designing and experimenting with new ones. Rather than packaging a number of different detection methods, our library focuses on implementing the core principles of our framework, then existing specific methods can be instantiated in a few lines of code and are readily available, as well as the possibility to benchmark new methods. We also package helpers to load existing weakly supervised datasets from the literature using a common interface in order to improve the reproducibility of experiments in the weakly supervised setup.

- **Empirical evaluation** – On a large benchmark of text and tabular datasets, we evaluate a series of detectors in different setups, where we vary the type of noise (weak supervision using labeling functions or uniform noise), the hyperparameter selection method (using a clean validation set or a noisy validation set), and the strategy for handling detected mislabeled examples (filtering or relabeling). We identify a number of practical questions for which our experiments provide new insights. We also share raw data for our experimental results as we believe they can be used to answer some future questions. Using our classifier-agnostic implementation, we propose the first comparison of different mislabeled detection methods on the exact same task and using the same features and base machine learning model.

## 2 Supervised learning and mislabeled examples: concepts and strategies

### 2.1 Definitions and problem statement

Training datasets that contain mislabeled examples are prevalent in real-world machine learning applications. **Our main aim is to detect these mislabeled examples.** A first challenge that we found while reviewing mislabeled detection methods is that it is difficult to come up with a general formal definition of what it means for an example to be mislabeled.

For some obvious cases, a human annotator can easily tell if an example is correctly labeled or mislabeled, but in some others the definition of mislabeled is more ambiguous such as when an instance can be considered to lie in 2 different classes. As an attempt at giving a precise definition, we distinguish between 2 theoretical frameworks in the next sections. We found no precedence of such an attempt.

#### 2.1.1 Deterministic case: the true concept is a function

We aim at estimating a (deterministic) function $f$ from an input space $\mathcal{X}$ to an output space $\mathcal{Y}$. Ideally we would observe a finite sample $\mathcal{D}_{\text{noiseless}}$ of $n$ examples and their corresponding labels $\{(x_i, y_i = f(x_i))\}_{1 \leq i \leq n}$ where the $x_i$ are sampled from an unknown distribution $\mathbb{P}(X)$ over the input space. Typically, in classification, $y_i$ is the class of the instance, whereas in regression, $y_i$ is a scalar value.

In real life, the data labeling process is often imperfect: our actual observed dataset $\mathcal{D}_{\text{train}}$ contains examples $\{(x_i, \tilde{y}_i)\}_{1 \leq i \leq n}$ that can undergo a corruption process and get a different label $\tilde{y}_i \neq y_i$. In this case, the definition is straightforward: an example is considered mislabeled if $\tilde{y}_i \neq f(x_i)$.

The approach of seeing machine learning as estimating an unknown function has permitted many theoretical and practical successes in the early days of *computational learning theory*. It, however, falls short of formalizing the more general case where each point in the input space corresponds to several different targets with non-zero (but possibly unbalanced) probability. This is the setup studied in *statistical learning theory* (see Luxburg & Schölkopf, 2011, for a concise but comprehensive overview of statistical learning theory).

#### 2.1.2 Stochastic case: the true concept is defined as a probability distribution

A more general case consists in defining the true underlying concept as a joint probability distribution $\mathbb{P}(X, Y) = \mathbb{P}(X)\mathbb{P}(Y|X)$. At fixed $x \in \mathcal{X}$, the mass of $\mathbb{P}(Y|X = x)$ is not necessarily concentrated into a single mode. Put differently, there are (possibly infinitely) many possible values for $y$ with non-zero probability. This happens for instance when $x$ does not contain all necessary information to predict $y$ and there remains some aleatoric uncertainty. Typically, in classification, we would like to learn an estimator that returns probabilities of belonging to each class $\hat{f}(x)_c = \mathbb{P}(Y = c|X = x)$, while in regression, it would return the expected value $\hat{f}(x) = \mathbb{E}[Y|X = x]$.

Similarly to the deterministic case, contrarily to the *independent and identically distributed* assumption often used in statistical learning theory, there is some discrepancy between the true concept that we want to learn, and the actual distribution of collected examples during the data labeling process: whereas we would ideally get a sample $\mathcal{D}_{\text{ideal}}$ of $n$ examples $\{(x_i, y_i)\}_{1 \leq i \leq n}$ from $\mathbb{P}(X, Y)$, we actually observe a training dataset $\mathcal{D}_{\text{train}}$ of examples $(x, \tilde{y})$ sampled from a corrupted distribution $\mathbb{P}(X, \tilde{Y})$.

In this case, the definition of a mislabeled example is ambiguous since we do not have a single ground truth value, but instead a probability distribution over the output space $\mathcal{Y}$ of many possible ones. This happens, for instance, when the true concept is modeled as a mixture of possible classes. As an example, suppose an input $x$ where $\mathbb{P}(Y|X = x)$ is a mixture of a majority class with probability 99% and a minority class with probability 1%, and suppose an example $(x, y)$ where $y$ is the minority class. Do we consider this example to be mislabeled?

In the rest, we suppose that we have access to a sensible threshold $\tau \in [0, 1]$: we consider examples $(x, y)$ with probability under the true concept $\mathbb{P}(Y = y|X = x) < \tau$ to be mislabeled and other examples to be genuine. Note that this is the true concept $\mathbb{P}(Y|X)$, which is unknown in general, not the output of an

estimator trained using a finite set of examples. This definition also covers the deterministic setting as a special case.

This choice is further motivated in data pipelines (section 2.6), where the output of a detection stage is fed to a filtering procedure that produces a dataset of most trusted examples used to train a machine learning estimator. In this case, we are often interested in recovering only the majority classes since they are also the most likely ones in our evaluation (test) set. However, we emphasize that in some other contexts, this definition might not be well suited, such as when we are more interested in predicting correctly for underrepresented instances in fairness-related tasks.

## 2.2 Detecting mislabeled examples using trust scores

Estimating the conditional probability is by itself a difficult problem, which e.g. requires proper calibration of machine learning models. Since we are only interested in splitting the dataset into genuine examples and mislabeled ones, it is sufficient to solve the relaxed problem of estimating a proxy of the conditional probability, hereafter called *trust score*.

**Definition 2.1** (Trust score). Any scoring function $s(x, \tilde{y})$ that is correlated with the conditional probability such that for any 2 examples $(x_1, \tilde{y}_1)$ and $(x_2, \tilde{y}_2)$, it preserves the ranking between conditional probabilities: $s(x_1, \tilde{y}_1) \leq s(x_2, \tilde{y}_2) \Leftrightarrow \mathbb{P}(Y = \tilde{y}_1 | X = x_1) \leq \mathbb{P}(Y = \tilde{y}_2 | X = x_2)$.

Equipped with this trust score, we can split the training dataset in 2 distinct parts: trusted examples with high trust scores and untrusted examples with low trust scores. For an ideal trust score method and threshold, the trusted and untrusted datasets are equal to the genuine and mislabeled examples sets. In section 3.2, we give a comprehensive review of model-probing methods for computing trust scores.

## 2.3 Assumptions regarding the noise generating process

A common approach to detect mislabeled examples within a dataset is to design detectors based on explicit assumptions regarding the structure of the underlying noise-generative process. A widely favored structure is known as the *noise transition matrix* denoted by $\mathbf{T}$. Here, $\forall (i, j) \in [\![1, \tilde{K}]\!] \times [\![1, K]\!]$, where $\tilde{K}$ is the number of noisy classes and $K$ the number of true classes, $\mathbf{T}_{i,j}$ represents the probability $\mathbb{P}(\tilde{Y} = i | Y = j)$ of an example from the class $j$ to have been assigned a noisy label from class $i$ (Van Rooyen & Williamson, 2017). This concept holds the advantage of generalizing over class-dependent label noise in a multi-class classification setting.

In the instance-dependent label noise scenario, the noise transition matrix becomes a function of the example $\mathbf{T}: \mathcal{X} \to \mathcal{M}_{\tilde{K}, K}(\mathbb{R})$, and in the uniform label noise scenario, it is reduced to a constant corresponding to the overall noise rate. A series of work has been dedicated to the estimation of the noise transition matrix for the class-dependent (Liu & Tao, 2015; Patrini et al., 2017; Xia et al., 2019; Yao et al., 2020) and instance-dependent (Xia et al., 2020; Yang et al., 2022) case.

However, additional assumptions are often necessary to estimate the noise transition matrix. Anchor point-based methods assume the existence and identifiability of high-confidence samples within the dataset (Liu & Tao, 2015; Patrini et al., 2017). Alternatively, some techniques require prior knowledge of class distributions or specific noise ratios (Wang et al., 2017), or they exploit the structural characteristics of the noise transition matrix, such as its tendency to form clusters in the feature space (Liu et al., 2023). Mixture proportion estimation approaches infer noise ratios or the noise transition matrix by assessing the contamination level of one class's feature distribution by others (Vandermeulen & Scott, 2016). However, these approaches presuppose that class distributions are mutually irreducible, implying distinct and non-overlapping patterns among classes (Scott, 2015).

In contrast, we assume no prior knowledge of the underlying noise structure in this paper. Instead, we chose to focus on detectors that do not *explicitly* depend on structural aspects of the noise-generating process, specifically focusing on the family of model-probing detectors. The success of these detectors relies on an *implicit* assumption concerning the base model's behavior, which should exhibit significant differences when applied to noisy versus clean data. In practice, model-probing detectors may fail in extreme cases where the

base model struggles to discern regularities in the data, e.g. due to excessive noise or an inductive bias in the base model that cannot capture the structure of the noise. In such situations, expecting them to identify mislabeled examples correctly may be unrealistic.

Furthermore, we note that more complex scenarios, such as concept drift (Lu et al., 2018) or data poisoning attacks (Tian et al., 2022), are out of the scope of this study but remain as open and interesting problems to tackle.

### 2.4 A taxonomy of data regions

In practical machine learning applications, we do not have exact knowledge of the true concept $\mathbb{P}(X, Y)$, but instead, we only have access to a limited sample of data points. Depending on the availability of data, by looking at the training set examples only, we distinguish between 4 cases (pictured in figure 1 for a 2-classes toy example):

Fig 1.(1) We have access to many examples and all are from the same class. In this case we can unambiguously consider this class as the true class: any example from another class lying in this region would be considered mislabeled.

Fig 1.(2) We have access to many examples, but they are equally spread into 2 different classes. In this case, no example from these 2 classes should be considered mislabeled.

Fig 1.(3) We only have access to a few examples but it looks like they all come from the same class. Does this mean that this is the true class, or just that data is too scarce in this region so we are just unsure about the true class?

Fig 1.(4) We only have access to a few examples, and they come from 2 different classes. Does that mean that we are close to a boundary of the true underlying concept with 2 separate regions from 2 different classes, or is it a region where examples from the 2 classes are sampled with equal probability from the true underlying concept?

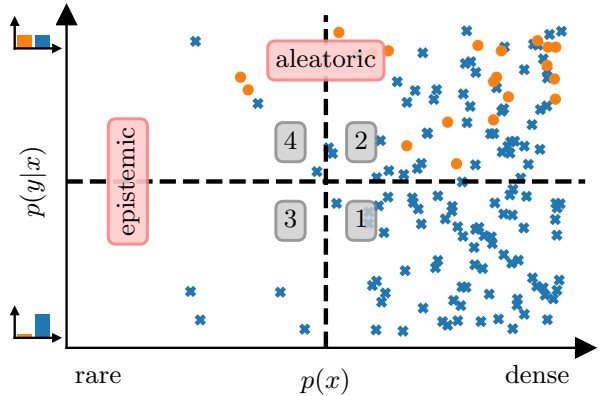

Figure 1: Illustration of a ground truth distribution $\mathbb{P}(X, Y)$ decomposed as $\mathbb{P}(X)$ on the y-axis and $\mathbb{P}(Y|X)$ on the x-axis and a sample of 100 data points from this distribution. In this toy distribution, we distinguish 4 different cases represented as 4 quadrants: **(1)** $\mathbb{P}(X)$ is dense, $\mathbb{P}(Y|X)$ is low entropy: we are pretty confident that the ground truth class is ✖ so any ● example would be mislabeled **(2)** $\mathbb{P}(X)$ is dense, $\mathbb{P}(Y|X)$ is high entropy, we cannot distinguish between classes ✖ and ● thus we would be unable to tell correctly labeled from mislabeled examples **(3)** $\mathbb{P}(X)$ is scattered, but $\mathbb{P}(Y|X)$ is low entropy so we can assume that the ground truth class is ✖ and any ● example should be deemed mislabeled **(4)** Since $\mathbb{P}(X)$ is scattered, it is more difficult to detect that $\mathbb{P}(Y|X)$ is high entropy by looking at the data only, it is likely that mislabeled detection methods would fail in the absence of further assumptions.

In the last case, by looking at the data alone, the main difficulty resides in distinguishing between rare examples which provide useful information regarding some specific part of the input space, and mislabeled examples. These rare useful examples are termed "exceptions" in Brodley & Friedl (1999), and are the ones that provide the last few percents in test accuracy in state-of-the-art classifiers (Feldman, 2020), or that carry most information regarding underrepresented subgroups in the population in fairness-related tasks (Liu et al., 2021).

A related concept is the distinction between aleatoric and epistemic uncertainty, where aleatoric uncertainty is high in regions where the true underlying concept $\mathbb{P}(Y|X)$ has high entropy, whereas epistemic uncertainty comes from a lack of knowledge due to not enough data points in specific regions (see Hüllermeier & Waegeman, 2021, for an introduction).

### 2.5 Use cases

Detecting mislabeled examples is of practical interest in real-world scenarios where obtaining the ground truth label of an example is an imperfect process. The causes of these imperfections are diverse and sometimes even intended in order to automatize as much as possible machine learning operations. We now highlight some imperfect labeling processes and the role that mislabeled example detection can play in these scenarios.

**Weak supervision**   Properly labeling an example sometimes requires a costly and non-scalable procedure prohibiting annotation of the whole training dataset. **Fraud detection** and **cyber security** are both fields where the labeling process requires scarce domain experts who need to conduct time-consuming forensic analysis. One of the most popular solutions is to distill expert knowledge into hand-engineered *labeling rules* to automatize and scale the labeling process to the entire training dataset. These rules form a new form of supervision called *weak supervision* (Ratner et al., 2016), and instead of the usual strong supervision, these labeling rules might produce incorrect labels that could harm the efficiency of machine learning model trained from these rules. Mislabeled example detection could potentially help experts design better rules to strengthen the weak supervision. Our experiments in section 5 include a benchmark of mislabeled detection methods in this setting.

**Crowd labeling**   When the labeling task is achievable by human annotators, outsourcing the labeling process to decentralized annotators is a common approach to annotate webscale datasets. This technique called *crowdsourced labeling* (Yuen et al., 2011) has been employed extensively in **computer vision** to create the first large image datasets such as ImageNet (Deng et al., 2009). Nowadays, it is employed to fine-tune **large language models** from human feedback (Ouyang et al., 2022; Bai et al., 2022). However, with crowdsourced labeling, the effectiveness of each annotator in following the labeling guidelines can vary and be unreliable. Mislabeled example detection provides a way to evaluate the annotators' ability to systematically follow the guidelines and even detect malicious annotators.

**Web scraping**   In order to quickly assemble supervised web-scale datasets free of human intervention, automatically crawling the web gathering data and labels from querying engines is a popular approach. *Web scraping* has been used in the **natural language processing** community to design sentiment analysis datasets (Maas et al., 2011) from movie reviews, or in the **computer vision** community to study label noise at scale (Xiao et al., 2015). As shown in the latter, the oracle used to label examples, here the search engine, sometimes diverges from the ground truth and provides noisy annotations. In the former, the data in itself was wrong because of human error, resulting in wrongly labeled data. More recently, *web scraping* has been extensively used to train **large language models**, in a self but still supervised fashion where the label to predict is the token next to the input sequence. However, using non-curated web data has been shown to severely hinder their capacities to produce non-toxic or hallucinations-free text (Wang et al., 2023). In these situations, mislabeled examples detection can provide insights on the quality of the data source used to automatically construct datasets.

In light of these scenarios, the data understanding part of the data mining methodology (Shearer, 2000) seems more prevalent than ever, caused by the ambition to automatize every part of the machine learning operations.

### 2.6 Fully automated learning in the presence of mislabeled examples: detect + handle strategies

Strategies to address learning with noisy labels include (Frénay & Verleysen, 2013): (i) using algorithms that are naturally robust to label noise; (ii) using algorithms that explicitly model label noise during training; (iii) assigning a trust score to each training example to then manually inspect low trust training examples, or use an automated downstream method that can leverage this additional metadata. All three families of methods are useful depending on the context. In the rest of the paper, we focus on the latter strategy: The detection of mislabeled instances is not only valuable for the sole purpose of curating a dataset that more accurately reflects the underlying concept, but it is also a critical preliminary step in a comprehensive pipeline, illustrated in figure 2. The ultimate goal is to train an estimator on a smaller dataset, free of mislabeled instances, with the expectation of getting more accurate predictions. It is achieved using a *detection* method that provides trust scores that are then fed to a *splitting* strategy that separates a *trusted* part of the training examples from *untrusted* ones. A final stage *handles* the 2 datasets in order to provide a trained estimator. Here we distinguish between 3 strategies:

**Filtering** The simplest approach for handling untrusted examples is to discard them from the training dataset, thereby training using *only trusted data*. This approach can also be found in the literature under the name of data cleaning or editing (Wilson & Martinez, 2000). From this perspective, it is not harmful to accidentally flag some correctly labeled examples as being untrustworthy as long as enough representative correctly labeled examples remain in the trusted dataset. To the contrary, some correctly labeled examples that lie in underrepresented regions might be accidentally flagged as mislabeled, which would degrade the performance of the final estimator in these regions of the feature space. Therefore the effectiveness of this approach is bound to the underlying detection method being able to distinguish between rare (thus difficult) and mislabeled examples and the precise tuning of the threshold used to split the dataset into trusted and untrusted examples.

**Semi-supervised learning** Although filtering is a direct approach, it may be overly dismissive of the information contained within untrusted data. Since this study is restricted to labeling noise, training instances are considered free of feature space noise. Thus, a more reasonable approach is to retain the instances while disregarding their labels, thus casting the handle step into a *semi-supervised* problem (Li et al., 2020).

This semi-supervised approach has the advantage of maintaining the entirety of the training dataset, thus preserving the original data distribution. However, it inherits the filtering method's intrinsic limitation of discarding some information from the untrustworthy examples (here, their labels), which could be partially correct or beneficial for the learning process.

**Biquality learning** The ideal approach would involve retaining the full training dataset while incorporating metadata about the quality of each example, allowing the learning algorithm to leverage this auxiliary information. A particular case of this scenario is called *biquality* data, where two datasets are available at training time, a trusted and an untrustworthy dataset, which falls under the biquality learning framework (Nodet et al., 2021). Algorithms within this framework are able to make more granular decisions regarding how untrusted examples are handled, potentially reweighting or relabeling individually each example. However, this approach assumes a high confidence in the trusted dataset: Any mislabeled example misclassified as trusted could significantly undermine the effectiveness of these algorithms.

## 3 Model-probing detection of mislabeled examples

### 3.1 Using trained models to detect mislabeled examples

Machine learning consists in identifying regularities in data sets and exploiting these regularities in order to predict the outcome on new examples. To the contrary, mislabeled examples are instead the ones that deviate from these regularities. The underlying concept behind model-probing mislabeled detection methods is that these irregular examples are treated somehow differently from genuine examples by the machine learning model. Informally, a good base model should find mislabeled examples *difficult* to learn and genuine examples *easier* to learn. In practice, quantifying how regular or irregular every example is, is done by probing trained

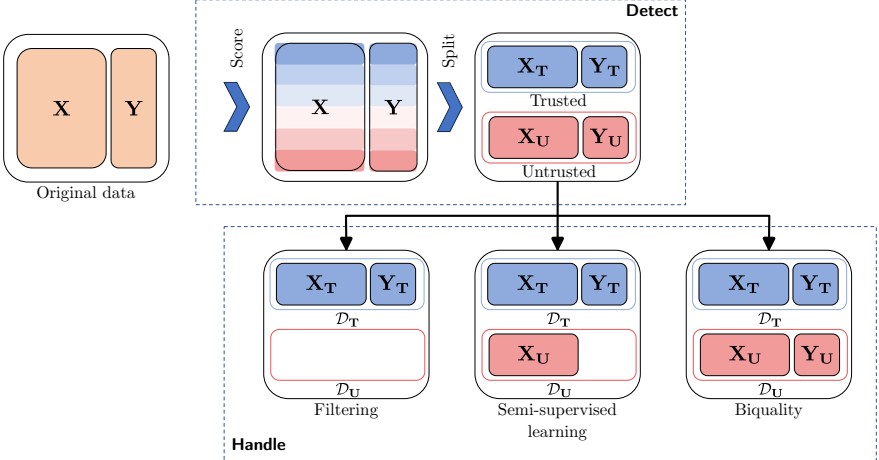

Figure 2: Data pipeline for different learning strategies when in the presence of labeling noise, where an intermediary step uses a detection method to assign trust scores to every example, then splits the dataset into a trusted and an untrusted part.

machine learning models (figure 3).To this end, similar to the fact that some machine learning methods may be more suited than others to some particular tasks (i.e. they get better generalization performance), accurately detecting mislabeled examples crucially depends on the choice of machine learning method and proper tuning of hyperparameters.

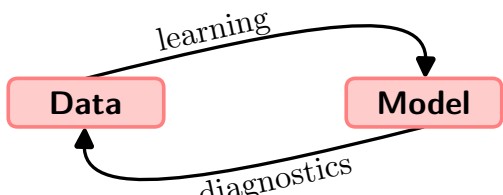

Figure 3: Whereas machine learning consists in choosing a model that best fits the data (from left to right), model-probing mislabeled detection methods follow the opposite direction and probe a trained model in order to give diagnostics on a set of examples (from right to left).

## 3.2 A general framework for model-probing methods with examples

We now discuss one of the contributions of our work, which is to provide a general framework that encompasses most methods for the identification of mislabeled examples, with a few exceptions discussed in subsequent sections. We also survey existing methods found in the literature and show that they fit in this framework. The framework is pictured in figure 4. The final outcome of these methods is a scalar trust score (defined in Section 2.1) for each example, used to rank examples from most likely to be mislabeled to most trusted. The framework is composed of 4 components that we shortly describe next. We then go in more depth in subsequent sections, where we show that most surveyed methods correspond to an instance of this framework using specific choices of each component.

**1/ base model:** All model-probing detection methods rely on fitting a machine learning model to the training set examples (or a subset thereof, depending on an optional ensemble strategy). This model should incorporate some robustness to mislabeled examples so that they are treated differently from genuine examples.

**2/ model probe:** We then probe[*] this model (possibly at different checkpoints during training, depending on the ensemble strategy), in order to score every example using a scalar metric used to discriminate between genuine and mislabeled examples.

**3/ ensemble strategy:** Optionally, the base model is trained multiple times using an ensemble strategy such as bootstrapping or boosting, where each learner of the ensemble provides a slightly different value of the probed scores.

**4/ aggregation method:** These scores are then aggregated to provide a single scalar trust score for each example.

As an example, the Area Under the Margin (AUM) method (Pleiss et al., 2020) fits in this framework by considering the base model to be a deep network, the probe to be the margin at every iteration during training, the ensemble strategy is the consecutive iterates, and the aggregation method is just the sum of these margins.

Table 1 summarizes all surveyed methods that fit in the framework. This unified picture suggests that we can automatically design new methods by simply replacing one or several of the components in already existing methods. This also advocates for transferring ideas developed for deep learning models to classical ones (section 4.3) and vice versa, where for instance the method of observing the fluctuations of the per-example accuracy as training progresses has been independently discovered in deep learning (Toneva et al., 2018) and with AdaBoost (Chen et al., 2022).

### 3.2.1 Base model

The base model is the core component of this framework, which is trained on the training examples (or a subset of the training examples depending on the ensemble strategy, see section 3.2.3) and then probed (section 3.2.2) to give a scalar score to every example. Intuitively, a good candidate base model should treat genuine and mislabeled examples differently, so that mislabeled examples are more difficult to learn than genuine ones. This form of robustness against learning mislabeled examples is for instance found in deep learning models (Arpit et al., 2017; Feldman & Zhang, 2020) where mislabeled examples have been shown to be handled differently (Krueger et al., 2017), in particular depending on the training regime (George et al., 2022), or in decision trees where mislabeled examples often end-up as a single instance of a different class in otherwise pure leaves. In general, every off-the-shelf machine learning method can be chosen, depending on the task (data and output types, and performance metric). Some detection methods however require specific characteristics for the base model, such as being differentiable with respect to their input (Agarwal et al., 2022) or their parameters (Koh & Liang, 2017).

The popularity of base models used for mislabeled examples is directly linked to the general popularity of machine learning models, where nearest neighbors methods used to be more popular in the early days of machine learning (Wilson, 1972; Tomek, 1976), then kernelized linear methods (Thongkam et al., 2008; Ekambaram et al., 2016) and more recently decision tree ensembles (Verbaeten & Van Assche, 2003; Chen et al., 2022). With the success of deep learning methods in image or text related tasks, new mislabeled detection methods have followed that exploit the specific features of neural networks such as their training dynamics (Toneva et al., 2018; Pleiss et al., 2020; Agarwal et al., 2022; Pruthi et al., 2020; Koh & Liang, 2017; Jiang et al., 2021), as well as classical machine learning methods on top of features extracted from deep networks representations (e.g. k-NNs in Bahri et al., 2020; Zhu et al., 2022; 2024).

As a **summary**, any supervised learning method can be used as a base model.

### 3.2.2 Model probe

Fitted models are then probed in order to get scalar scores that are used to discriminate between genuine and mislabeled examples (figure 3). Probes produce intermediate values that are then aggregated (section

---

[*]We use the word *probe* throughout which is generic enough to encompass a variety of different methods that follow the same purpose of scoring each example by means of some measurement on a trained model.

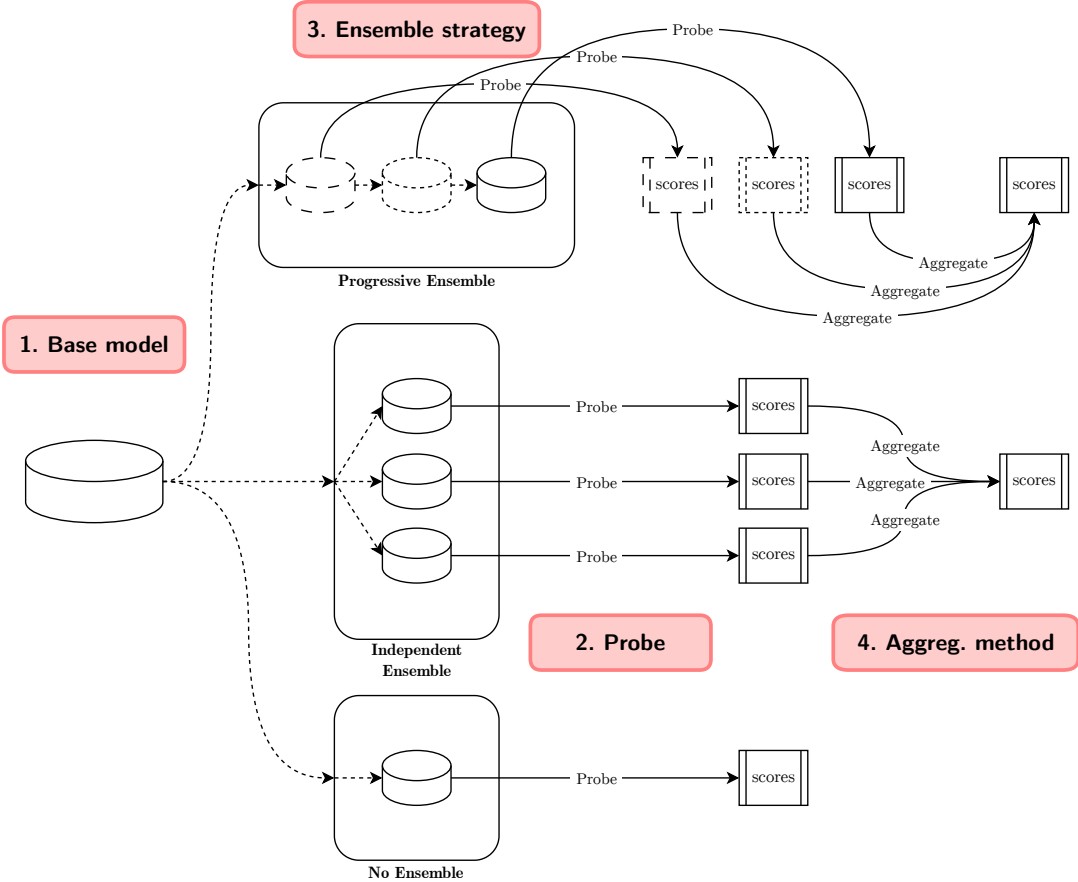

Figure 4: Schematic summary of model-probing detection methods, with their 4 components.

3.2.4) to a single scalar trust score for each example. The most naive way of probing a model is simply to use its prediction, but it can also be done using more convoluted methods, some that are similar to the concept of uncertainty in active learning (Settles, 2011), some others which are more specific to the machine learning model used such as gradient of logits with respect to input pixels in Agarwal et al. (2022).

An early series of detection methods simply use the **predicted class** by comparing the prediction given by neighbor examples with a k-NN classifier (Wilson, 1972; Tomek, 1976; Brodley & Friedl, 1999) or a SVM (Segata et al., 2010; Thongkam et al., 2008) and flag examples as mislabeled if the prediction does not match the dataset label. Following a different motivation but similarly using the predicted class as the probe, forget scores (Toneva et al., 2018) in deep learning and fluctuation scores (Chen et al., 2022) in AdaBoost measure how much the per-example accuracy oscillates as training progresses, where more oscillations indicate a more difficult thus potentially mislabeled example.

AdaBoost **example weights** (Verbaeten & Van Assche, 2003) have been used as probes, with larger weights indicating more difficult examples. This is very similar to the method of **small loss** popularized by a series of methods in deep learning (Amiri et al., 2018; Jiang et al., 2018). Here, examples with small loss are considered genuine and larger loss are a sign of suspicious examples. In the context of learning with the cross-entropy loss, this is also similar to using the **margin** (Dligach & Palmer, 2011; Pleiss et al., 2020), where a small or negative margin indicates a potential mislabeled example, or to consider **support vector** examples (Ekambaram et al., 2016) as suspicious as these are the examples closer to the decision boundary in support vector machines. In Paul et al. (2021), the $\ell_2$ **distance** between the one-hot encoded label and the softmax output of a deep network defines the EL2N score which is generally higher for noise instances.

Following the principle that perfectly fitting mislabeled examples would require more complex models, a family of methods estimates the additional **complexity** of the base model required to learn an example compared to removing the example from the training set (Gamberger et al., 2000). A proxy measure for complexity is used in Chen et al. (2022) as the **number of weak learners** required to learn the label. In Ma et al. (2018), a high value of the **local intrinsic dimensionality** of the last layer representation for some examples as training progresses indicates the need for a higher compression in order to learn these examples. In Baldock et al. (2021) in deep learning, classifiers are trained using the representations extracted from layers of varying depth, the smallest depth able to correctly classify a data point measures how easy it is, thus higher **prediction depths** correspond to potential mislabeled examples. With a different motivation, Agarwal et al. (2022) propose to measure the **gradient** of the target logit of individual dataset examples with respect to the input space, which is a measure of the smoothness of the prediction function. This builds on the idea that perfectly fitting a mislabeled example in a region with otherwise genuine examples requires a very localized spike in the decision function, which will be reflected in a larger gradient.

A popular tool in statistics, **influence functions** (Hampel, 1974) have been adapted to machine learning as a diagnostic tool to estimate the effect of adding or removing an example in the training dataset. Informally, they are an estimate of the effect of an infinitesimal change in the weight given to a datapoint, to some value of interest, such as the parameters of the optimal model learned from this new weighting of examples, or the prediction on other examples (also see Cook, 1977, for closed-form expressions of several variants applied to linear models). In deep learning, they are estimated using linearization of the predictor near an optimum (Koh & Liang, 2017; Barshan et al., 2020; Kong et al., 2021; Bae et al., 2022). Self-influence is defined as the influence of an example on its own prediction. High values of self-influence indicate that the prediction on an example is only influenced by itself, whereas a low self-influence is a hint that other examples carry similar features and target, indicating a more trustworthy example. In TracIn (Pruthi et al., 2020), the influence of examples is instead estimated using the gradient of the per-example loss at several checkpoints during training, also similar to GraNd (Paul et al., 2021) which use the averaged gradient norm as we vary the initial parameters of a neural network. Inspired by the representer theorem for functions obtained by minimizing the empirical risk over a reproducing kernel Hilbert space (RKHS), **representer values** (Yeh et al., 2018) aim at explaining a trained deep model's prediction by means of the contribution from each training point. Similar to self-influence values, these coefficients can be used to diagnose mislabeled examples.

In Sedova et al. (2023), the **cosine similarity between the gradients of the loss** of individual examples, and the average gradient of the loss estimated on a minibatch that does not contain these examples, is used as an indication that this example disagrees with other examples, in that it would push the learned model towards a different direction than that of the majority of examples. Low or negative cosine similarity indicates potential mislabeled examples.

While most probes discussed so far are specialized to classification tasks, the same model-probing framework also applies when dealing with regression by just choosing appropriate probes. This is e.g. done in Zhou et al. (2023) where the model is probed using the $\ell_1$ **distance** between the prediction of the trained model and the dataset target.

Designing new ways of probing trained models can lead to improved detection of mislabeled examples. This is for instance explored in Kuan & Mueller (2022).

**Summary of probes:** predicted class, boosting weights, loss, margin, $\ell_2$ distance one-hot, support vectors, number of weak learners, local intrinsic dimensionality, prediction depth, input/output gradient, influence function, representer values, $\ell_1$ distance, similarity between gradients.

### 3.2.3 Ensemble strategy

Whereas some detection methods only rely on a point estimate (e.g. Wilson, 1972; Segata et al., 2010), mislabeled detection methods can be improved by leveraging an ensemble strategy (Verbaeten & Van Assche, 2003). We distinguish between the *independent* and *progressive* families of ensemble methods, that differ by the nature of the members of the ensemble, and how we leverage the differences in weak learners in the context of detecting mislabeled examples.

Independent ensembles are built using **bootstrapping**, **cross-validation** or **leave-one-out**, where models are trained on subsets of the original training set. This produces a variety of measurements of the probe. In these methods, all weak learners are trained on *independently* drawn subsamples of the training set, they can be considered to be on equal terms. The disagreement or inconsistency of different models within the ensemble can be leveraged as an indication of a potential mislabeled example (Jiang et al., 2021; Northcutt et al., 2021a). Instead of varying the subset of examples on which ensemble members are trained, it is also possible to train using **different models** (Smith & Martinez, 2014), and leverage their diversity.

By contrast, in progressive ensembles, there is a natural ranking between weak learners that come from the consecutive iterations. In **boosting**, a predictor is *progressively* built as a sum of weak learners, each of which is learned from the prediction of the previous boosting iteration. In the context of the detection of mislabeled examples, boosting models can be probed at every iteration during training, which produces a series of different values (Chen et al., 2022). **Deep learning** models are also trained in an iterative fashion (Pleiss et al., 2020), where each step in the parameter space incurs a change in function space that can be assimilated to a weak learner[*]. Because of this analogy, methods designed to work with deep networks can be readily adapted to boosted models (section 4.3). In progressive ensemble strategies, early iterations learn a prominent pattern (ideally corresponding to the clean examples) whereas late iterations are required to learn spiked decision boundaries as well as mislabeled examples. In this case, the complexity of the decision function learned increases as training progresses, which can be leveraged as an extra signal to detect clean and mislabeled examples.

**Summary of ensemble strategies:** bootstrapping, cross-validation, leave-one-out, different models, boosting/deep learning.

### 3.2.4 Aggregation method

We obtain a series of probe scores from each model of the ensemble. In order to summarize them to a single scalar trust score, we now need to aggregate all these measurements. This is achieved by choosing an aggregation method. The simplest one is just to **average** the probed scores (or equivalently, take their **sum**) as done e.g. in Northcutt et al. (2021a); Pleiss et al. (2020); Pruthi et al. (2020), or compute a **majority vote** or **consensus** as in Guan et al. (2011); Verbaeten & Van Assche (2003).

A natural extension is instead to measure some form of spread of probed scores, with a large spread indicating that ensemble members disagree thus a potential mislabeled example. This is e.g. achieved using the $\ell_2$ **variance** in Agarwal et al. (2022); Seedat et al. (2022), also reminiscent of the idea of uncertainty in active learning (Chang et al., 2017). Going a step further, the **difference** in probe scores between members of the ensemble that include a given example in their training subset and members that do not (e.g., van Halteren, 2000) can also be leveraged, where a larger difference suggests a potential mislabeled example, whereas a prototypical example will probably have less influence on the predictor since similar examples are likely included in the training subset. For instance C-scores (Jiang et al., 2021) are a measure of how likely a datapoint is to be correctly classified by a model trained on a subset that does not include it. Similarly, **DataShapley values** (Ghorbani & Zou, 2019; Kwon & Zou, 2022) measure the individual value of each example on a utility function such as the test loss, with a negative value indicating a potentially mislabeled example.

For progressive ensemble strategies, there is a natural ordering of the members of the ensemble, which are the consecutive iterations of the algorithm. This is exploited by several techniques. Forget scores in deep learning (Toneva et al., 2018) and **fluctuation** (Chen et al., 2022) in AdaBoost count how many times the per-example accuracy between consecutive iterations changed from not predicting the dataset label to correctly predicting it, with larger values indicating that the example is more difficult to learn. In Cordeiro et al. (2023), examples are considered clean if their individual loss is smaller than a threshold $\tau$ for $\zeta$ epochs in a row where $\tau$ and $\zeta$ are 2 hyperparameters, and in Yuan et al. (2023), trustworthy examples are those

---

[*]At the end of training, $f_{\mathbf{w}_T} = f_{\mathbf{w}_0} + \sum_{t=1}^{T} \underbrace{f_{\mathbf{w}_t} - f_{\mathbf{w}_{t-1}}}_{:=h_t}$ can be viewed as an ensemble of weak learners $\{h_t\}_{t\in[\![1,T]\!]}$ stemming from parameters updates $\mathbf{w}_t - \mathbf{w}_{t-1}$.

| Base model | Probe | Ensemble strategy | Aggregation | |
|---|---|---|---|---|
| k-NN | accuracy | leave-one-out | OOB value | Wilson (1972) |
| k-NN | accuracy | no ensemble | n/a | Tomek (1976) |
| various | accuracy | bootstrapping | majority vote | Brodley & Friedl (1999) |
| AdaBoost | example weights | no ensemble | n/a | Verbaeten & Van Assche (2003) |
| SVM | accuracy | no ensemble | n/a | Thongkam et al. (2008) |
| Local SVM | accuracy | no ensemble | n/a | Segata et al. (2010) |
| MaxEnt | margin | no ensemble | n/a | Dligach & Palmer (2011) |
| various | self confidence | different models | sum | Smith & Martinez (2014) |
| SVC | support vectors | no ensemble | n/a | Ekambaram et al. (2016) |
| deep network | influence | no ensemble | n/a | Koh & Liang (2017) |
| deep network | accuracy | learning iterations | change count | Toneva et al. (2018) |
| deep network | loss | no ensemble | n/a | Amiri et al. (2018) |
| deep network | local intrinsic dim. | no ensemble | n/a | Ma et al. (2018) |
| deep network | representer value | no ensemble | n/a | Yeh et al. (2018) |
| deep network | loss | no ensemble | n/a | Jiang et al. (2018) |
| deep network | margin | learning iterations | sum | Pleiss et al. (2020) |
| deep network | loss gradient | learning iterations | sum | Pruthi et al. (2020) |
| k-NN | accuracy | no ensemble | n/a | Bahri et al. (2020) |
| deep network | $\ell_2$ distance one-hot | no ensemble | n/a | Paul et al. (2021) |
| deep network | accuracy | bootstrapping | sum | Jiang et al. (2021) |
| deep network | prediction depth | no ensemble | n/a | Baldock et al. (2021) |
| deep network | self confidence | bootstrapping | mean | Northcutt et al. (2021a) |
| deep network | influence | no ensemble | n/a | Kong et al. (2021) |
| deep network | input/output gradient | learning iterations | variance | Agarwal et al. (2022) |
| decision stump | accuracy | boosting iterations | change count | Chen et al. (2022) |
| AdaBoost | # weak learners | no ensemble | n/a | Chen et al. (2022) |
| k-NN | accuracy | no ensemble | n/a | Zhu et al. (2022) |
| various | $\ell_1$ distance | cross-validation | OOB value | Zhou et al. (2023) |

Table 1: Taxonomy of model-probing detection methods

that are correctly classified for $k$ epochs in a row earlier during training, where $k$ is also a hyperparameter. The $\ell_2$ variance of the input/output gradient of individual examples across iterations is used in Agarwal et al. (2022), with the idea that the training dynamics of a neural network will fluctuate around mislabeled examples as it will require a decision function that spikes around a single example in an otherwise uniform region.

Instead of aggregating different scores from different members of an ensemble, it is also possible to compute $k$ different probes for a single trained model, as done in Lu et al. (2023) for $k = 2$. Here, the scores are aggregated in a single trust score using a Gaussian mixture model (GMM), but in general, we could expect any $k$-dimensional **clustering** method to work.

**Summary of aggregation methods**   average/sum, majority vote, consensus, variance, difference in vs out, DataShapley, difference between iterates, stability for $\zeta$ epochs in a row, clustering in higher dimension

### 3.3   Bag of (clever) tricks

The framework proposed in section 3.2 is the backbone for many reviewed detection methods. In addition, we now survey some additional techniques proposed in the literature, which we consider as plugins that aim at solving specific problems that arise when dealing with mislabeled examples. These methods are applied on top of the framework and are agnostic to the model-probing detection strategy.

### 3.3.1 Iterative refinement

The pipeline for learning in the presence of mislabeled examples detailed in section 2.6 is a 2-stage approach with detect and handle stages applied once. A natural way to improve its efficiency is to do **many-passes** over the training examples by probing base models fitted on a sequence of refined datasets iteratively, where the base models of iteration $T$ are trained using clean examples only filtered by iteration $T - 1$ (Tomek, 1976; Chen et al., 2019).

The iterative refinement approach can be integrated directly into the training procedure of the machine learning model. As training progresses, the sets of beneficial and detrimental examples may change. By incorporating the detection stage into each iteration of the training procedure, the model can be updated incrementally with the suitable refined dataset given its current progress in its curriculum (Sedova et al., 2023).

### 3.3.2 Surely mislabeled pseudo-class

The trust scores generated by most detection methods are often on an arbitrary scale. Splitting a training set into a trusted and untrusted part thus requires a carefully chosen threshold. This threshold depends on the detection method (and scale of the trust scores), as well as on the noise level and structure of noise. When used in a detect + filter pipeline, choosing an appropriate value of the threshold is of crucial importance. It can be achieved by treating it as a hyperparameter in a cross-validation setup (which likely requires a noise-free validation set, see discussion in section 5.3).

Alternatively, Pleiss et al. (2020) proposed to artificially introduce examples that are purposely mislabeled (we assign them a wrong class) and measure their trust scores so that we get a distribution of trust scores corresponding to surely mislabeled examples. This is achieved by introducing an extra *pseudo-class* and assigning it to a random subset of the training data. It, however, assumes that the noise introduced by this additional class has similar properties as the noise in the original data.

### 3.3.3 Class-balancing mechanisms

A limitation of detection methods arises when dealing with class-imbalanced datasets. Given that minority class examples are scarce, they are often more difficult to correctly predict than other examples, thus they are more prone to being flagged as potentially mislabeled examples by most detection methods. In the meantime, they are often the most useful ones given their scarcity: we would not be able to correctly predict the minority class if there were to few examples from the minority class in the training data.

In order to alleviate this issue, a reasonable approach is to detect mislabeled examples in a **one vs. rest** fashion by selecting trusted examples class per class (Northcutt et al., 2021a; Wang et al., 2022; Karim et al., 2022).

An alternative approach is to **normalize** scores across classes allowing the splitting step to be done parsimoniously for all classes, which can be done by a simple scaling (Kim et al., 2023), or done under the peered prediction framework (Miller et al., 2005) using peer examples (Liu & Guo, 2020; Cheng et al., 2020).

An alternative could revolve around the **calibration** of the trust scores compared to the true conditional probability, which remains an under-explored area. The only works we are aware of that attempt to tackle this problem proposes to adjust the predicted probabilities of training examples by the average predicted probability for each class while probing the model (Northcutt et al., 2021a; Kuan & Mueller, 2022).

### 3.3.4 Reducing epistemic uncertainty

Another interpretation of the problem of differentiating hard but clean examples from noisy examples is through the field of uncertainty quantification, specifically in the distinction between epistemic and aleatoric uncertainty (Hüllermeier & Waegeman, 2021). Epistemic uncertainty corresponds to uncertain predictions of a base model that can be reduced with more training data (rare and hard examples). Aleatoric uncertainty corresponds to irreducible uncertainty, where the information from the features of a sample alone cannot predict its label (e.g. when the true underlying concept is a mix of several classes).

When probing the base model, both of these categories of examples will be assigned a low trust score (Hooker et al., 2019). To better differentiate these types of uncertainty, **data augmentation** can be used to artificially create more training data (D'souza et al., 2021). This way, trust scores of examples with epistemic uncertainty will increase, while trust scores of examples with aleatoric uncertainty will remain the same.

An alternative strategy involves designing detectors that combine probes that respond differently to aleatoric and epistemic uncertainty. In Kuan & Mueller (2022); Zhou et al. (2023), it is done by re-weighting label-noise probes (respectively the self-confidence and the $\ell_1$ distance) by out-of-distributions probes (respectively the prediction entropy and the prediction variance). In Lu et al. (2023), a label-noise probe and an out-of-distribution probe are computed separately and aggregated through clustering.

Furthermore, framing the problem of mislabeled examples detection as an outlier detection task is another way to disambiguate epistemic and aleatoric uncertainty, yet out of scope of the model-probing framework (see section 6 for a more thorough discussion regarding detection of outliers).

## 4 Library

To further emphasize that the proposed framework in section 3.2 is not only of theoretical interest but also of practical one, we now present another important contribution of our work in the form of a Python library that materializes the 4 components (base model, probe, ensemble strategy, aggregation method) of the framework into a modular approach with 4 blocks that can readily be customized. Implementing an existing method of the literature from table 1 then amounts to just specifying the value of each column of the table, and we can invent new methods as new combinations of already existing components.

### 4.1 Detection of mislabeled examples by computing trust scores

The core of the library is a versatile `ModelProbingDetector` object that uses 4 arguments:

- a `BaseModel` which can be any estimator using scikit-learn's API (Pedregosa et al., 2011),

- an `EnsembleStrategy` that defines the logic on how to fit and probe the base model,

- the `probe` that returns a score for every training example given a fitted model,

- and the `aggregator` that defines how we aggregate scores over multiple probes and/or multiple fitted models.

Trust scores for training examples `X` and corresponding, possibly corrupted, labels `y` are then computed by calling the `.trust_scores(X,y)` method of `ModelProbingDetector` which closely follows scikit-learn's design so that most machine learning practitioners should already feel familiar. The method returns a scalar trust score for each example.

For example, the `AreaUnderMargin` detection method (AUM, Pleiss et al., 2020) can be defined to work with gradient boosted trees using scikit-learn's implementation `GBT` with the following code snippet:

```
AreaUnderMargin = ModelProbingDetector(
    base_model=GBM(),
    ensemble=ProgressiveEnsemble(),
    probe="margin",
    aggregate="sum",
)
```

Figure 5: This code reads "*consider a gradient boosted tree model (GBM) as a progressive ensemble, compute margins for all examples at every iteration during training, sum them up to obtain scalar trust scores*".

Noteworthy, we can readily perform ablation studies with respect to any of the 4 components by keeping all 3 others fixed. For instance, implementing the same detector but using iterates of a logistic regression model trained with gradient descent can be done by replacing `GBM` with scikit-learn's `LogisticRegression` estimator. Similarly, instead of using the margin, we could design a different probe for a specific need, and

e.g. imagine an alternative implementation of AUM for regression that would instead use the $\ell_2$ distance to the target as its probe (see section D.1).

The library comes with a series of helpers for defining most detectors found in the literature, so that all detectors benchmarked in section 5 are readily available with a simple Python import.

## 4.2 A versatile API

We designed our library so that we can easily extend it with new ideas (e.g. new ways of probing a base model), as long as each component follows the following block contracts:

- The `BaseModel` contract follows the widely used API of scikit-learn's estimators (Buitinck et al., 2013). It allows using scikit-learn's suite of already implemented estimators, as well as estimators from other libraries that follow the same widely used API.

- The `EnsembleStrategy` contract is a single method `.probe_model(base_model, X, y, probe)` that takes as an input a non-initialized base model, the features, the labels, and the probing method, and outputs the computed probes as an iterator of length `n_models` yielding NumPy arrays of shape `(n_samples, n_probes)`, and potential metadata, such as an iterator of boolean masks indicating if a sample was part of the training set of the probed model (e.g. in the case of bootstrapping it allows to distinguish in-the-bag from out-of-bag examples).

- The `probe` contract is a callable that takes as an input a fitted model, features, and labels of training samples and outputs a one-dimensional NumPy array of length `n_samples`.

- The `aggregator` is a callable that defines how we summarize a series of probe scores viewed as an iterator yielding NumPy arrays of shape `(n_samples, n_probes)` to a one-dimension NumPy array of trust scores of length `n_samples`.

Thanks to the modularity of the API as well as the ease of adding new components, exploring new detectors uncovered in table 1 (unknown region of a 4D cube) is as easy as changing one string when instantiating a `ModelProbingDetector`. We hope that our library can foster the design and understanding of future mislabeled detection methods.

## 4.3 A common API for progressive ensembles

We propose a novel API to unify all incremental machine learning approaches into a single contract named `staged_fit` that produces a *stream* of machine learning models from a dataset. As streams are lazy data structures, it allows flexible implementations of this contract for different families of machine learning models. For deep networks, to reduce memory cost, only a single model is kept in memory, and the network is trained incrementally between each iteration. For gradient boosting machines, all trees are trained at once and then copied and dispatched into smaller GBT s for each iteration. Moreover, the concept of iteration can be changed dynamically and independently for different model families. The provided implementation for deep networks uses an epoch as the reference for an iteration, but a batch version could be used instead. As long as a notion of increment in complexity exists for a family of machine learning models, a `staged_fit` can be implemented. For example, decision trees are treated as progressive models, from decision stumps to fully grown trees with pure leaves.

## 4.4 Full pipelines

In addition to the detection API using `ModelProbingDetector` objects, we also provide a way of building full pipelines following detect + handle strategies as described in section 2. `Splitter` objects define strategies to split a training set in a trusted and untrusted part using the trust scores (e.g. using a specified threshold or by keeping a specified top quantile of the trust scores). `Handler` objects implement connectors to learning strategies such as filtering, semi-supervised learning, or biquality learning so that all necessary tools to create fully automated data pipelines are readily available.

# 5 Benchmarks

A classical approach to benchmarking detection methods is to evaluate them on synthetic tasks where noisy labels are injected artificially into otherwise clean datasets. It allows us to conduct a post-mortem analysis on the accuracy of detecting mislabeled examples since both the noisy and ground truth labels of all examples are known. Yet, using only synthetic corruptions in experimental protocols might lead to wrong conclusions on the actual performance of detection methods as they might not be representative of real-world corruptions. In our benchmark, we use text and tabular datasets with noisy labels generated from imperfect labeling rules, corresponding to more realistic scenarios, thanks to the growing availability of such datasets. On these tasks, we evaluated multiple surveyed detection methods on different criteria, most notably in the case of the fully automated weakly-supervised pipeline described in section 2.6.

Overall, the purpose of this benchmark is not to provide a definitive ranking between detection methods. As we surveyed a large spectrum of existing and adapted methods, we could not exhaustively fine-tune each method individually, so it is likely that some methods were not evaluated at their best capacity. Rather, this benchmark serves to highlight a few recommendations for practitioners, as it provides data for a meta-analysis on a large number of actual datasets.

## 5.1 Benchmark design

We now outline the features of the benchmark. A more detailed presentation with all specifics is included in appendix B.

**Tabular data**  We choose to put the emphasis of our benchmark on tabular tasks. Arguably (Grinsztajn et al., 2022), these tasks do not benefit from the representation learning properties of deep models, thus our evaluation only relies on the regularization properties of the machine learning models used instead of how good they are at learning useful representations. Tabular data tasks also often include ambiguous mixing regions, where the true underlying concept is a mix of several classes (Seedat et al., 2022). For illustration, we also include an experiment on an image dataset in appendix D.2.

**Datasets**  We use weakly supervised datasets from the Wrench benchmark (Zhang et al., 2021), supplemented by other datasets from the literature (Rühling Cachay et al., 2021; Hedderich et al., 2020), for which we have access to weak labels from a set of automatic labeling rules as well as ground truth labels. A summary of tasks statistics is available in table 2.

**Sources of noise in benchmarked datasets**  We experiment with the following setups :

- Artificial uniform noise: with probability 30%, an example is assigned a random label uniformly between existing classes, independently of its feature or true class. This creates a dataset with Completely At Random (NCAR) noise. In practice, this form of noise is often used when experimenting since it can easily be artificially introduced in existing noise free datasets. It is, however, very different (and simpler to deal with) from the actual structure of noise encountered in real-world datasets.

- Imperfect labeling rules: a set of automatic labeling rules are applied to every example and then aggregated as a single label. These labeling rules are imperfect, rendering the assigned labels noisy. In this case, noisy examples are more frequent in regions that are not correctly covered by the labeling rules or when several labeling rules disagree, thus the probability for examples to be mislabeled depends on their features $x$. The structure of the noise is Not At Random (NNAR). Some examples might not be covered by any labeling rule[*]. This more general form of noise is often more representative of actual use cases, but also more difficult to tackle.

---

[*]In these experiments, we chose to exclude examples that are not covered by any labeling rules from the training set. Alternatively, one could assign them a random label, but it would likely be a noisy one.

Table 2: Datasets used to benchmark detectors. Columns: dataset size $n$, number of raw features $d$, number of encoded features $\phi(d)$, number of classes $K$, histogram of class priors $p(y)$, number of labeling rules LRs, noise transition matrix $\mathbf{T}$, noise ratio $p(\tilde{y} \neq y)$, percentage of examples for which at least one labeling rule gave a label *coverage*.

| Benchmark | Modality | Dataset | $n$ | $d$ | $\phi(d)$ | $K$ | $p(y)$ | LRs | $\mathbf{T}$ | $p(\tilde{y} \neq y)$ | coverage |
|---|---|---|---|---|---|---|---|---|---|---|---|
| waln | text | hausa | 2.92K | 4.82K | 750 | 5 | | 18.7K | | 0.50 | 97% |
| | | yoruba | 1.91K | 6.95K | 539 | 7 | | 20.3K | | 0.40 | 93% |
| weasel | text | amazon | 200K | 160K | 3.68K | 2 | | 175 | | 0.25 | 65% |
| | | professor-teacher | 24.6K | 113K | 4.06K | 2 | | 99 | | 0.18 | 81% |
| wrench | tabular | bank-marketing | 45.2K | 16 | 78 | 2 | | 20 | | 0.26 | 93% |
| | | basketball | 20.3K | 2.05K | 2.05K | 2 | | 4 | | 0.25 | 100% |
| | | bioresponse | 3.75K | 1.78K | 10.4K | 2 | | 20 | | 0.46 | 99% |
| | | census | 31.9K | 105 | 105 | 2 | | 83 | | 0.19 | 99% |
| | | commercial | 81.1K | 2.05K | 2.05K | 2 | | 4 | | 0.10 | 100% |
| | | mushroom | 8.12K | 22 | 108 | 2 | | 20 | | 0.13 | 99% |
| | | phishing | 11.1K | 30 | 46 | 2 | | 15 | | 0.21 | 97% |
| | | spambase | 4.6K | 57 | 57 | 2 | | 15 | | 0.25 | 97% |
| | | tennis | 8.8K | 2.05K | 2.05K | 2 | | 6 | | 0.13 | 100% |
| | text | agnews | 120K | 145K | 3.36K | 4 | | 9 | | 0.19 | 69% |
| | | imdb | 25K | 74K | 10.1K | 2 | | 5 | | 0.26 | 87% |
| | | sms | 5.57K | 13.5K | 1.37K | 2 | | 73 | | 0.03 | 40% |
| | | trec | 6.03K | 9.25K | 946 | 6 | | 68 | | 0.46 | 95% |
| | | yelp | 38K | 200K | 5.35K | 2 | | 8 | | 0.28 | 82% |
| | | youtube | 2.06K | 7.08K | 423 | 2 | | 10 | | 0.15 | 89% |

**Detection methods** We evaluate several detection methods surveyed in section 3, choosing different detectors with the right diversity of components: Some originate from the deep learning literature, using progressive ensemble: Variance of Gradients (VoG, Agarwal et al., 2022), Area Under the Margin (AUM, Pleiss et al., 2020), Forget Scores (Toneva et al., 2018), TracIn (Pruthi et al., 2020), Small Losses (Amiri et al., 2018; Jiang et al., 2018) and AGRA (Sedova et al., 2023), others are not specific to deep learning: Consensus Consistency (Jiang et al., 2021) and CleanLab (Northcutt et al., 2021a), finally some methods come from the influence literature in linear models: Self-Influence (Koh & Liang, 2017) and Self-Representer (Yeh et al., 2018). Their respective position in our framework is described in table 1.

**Evaluation of detection methods** Evaluating detection methods is often task specific. We choose to evaluate surveyed methods using the following criteria:

1. Predictive power of the trust scores to detect mislabeled examples. We use the Area under the Receiver Operator Curve (AUROC) as a ranking quality metric.

2. Representativeness of the filtered dataset. We use class-balance as a proxy of representativeness. To measure class-balance in multi-class classification, we use the ratio of the prior of the minority class over the prior of the majority class.

3. Performance in a fully automated weakly supervised pipeline with no additional supervision. We use the test loss of an estimator trained after filtering of the less trusted examples given by a method's trust scores (detect + filter).

4. Performance in a semi-automated pipeline with additional supervision. We use the test loss of an estimator trained after relabeling of the 10% less trusted examples given by a method's trust scores (detect + relabel).

These pipelines provide a practical method for evaluating mislabeled detectors: an efficient detector must at least be able to somewhat distinguish between genuine and mislabeled examples, but it would be too restrictive to assess the quality of a detector by only measuring its accuracy. In particular, some examples are more important than others (e.g. because they are rare instances, or they belong to a minority class or an underrepresented pattern), and this should be reflected in the evaluation of detection methods. Instead, we employ these fully automated pipelines as a methodology for the evaluation of detection methods, whereby some classification metric (here the test loss) is measured for classifiers trained on filtered datasets. This approach allows for the disparate value of different examples to be taken into account: two detectors that have the same error rate but on different instances will result in different classifiers trained on filtered data.

**Hyperparameters**  The performance of mislabeled detection pipelines depends on the value of the following hyperparameters:

- Hyperparameters of the detection method (This includes hyperparameters of the base model such as e.g. the $\ell_2$ regularization coefficient in logistic regression, as well as the hyperparameters of the probe, ensemble and aggregation strategies).

- Hyperparameters of the final estimator.

- Threshold for splitting the training dataset between trusted and untrusted examples.

Hyperparameters or the base model of the detector, as well as hyperparameters of the final estimator are sampled by random search (12 times × 12 times). For each sampled couple of hyperparameters, threshold values are then chosen from the grid $\{0, 0.1, 0.2, 0.3, 0.4, 0.5, 0.6, 0.7, 0.8, 0.9\}$.

*Approximately 3 millions models have been trained in the making of this benchmark.*

**Choice of hyperparameter values**  Hyperparameter tuning is done using cross-validation, where an holdout split of the training dataset is kept apart from training examples, and used only to compute an estimate of the test loss. We distinguish between the following cases:

- Noisy: the validation set follows the same distribution as the training set. In particular, it contains potential noisy examples

- Noise free: the validation set only contains clean examples. An example use-case is when an additional effort is made on labeling of the validation set: since it is typically smaller than the training set, it is not prohibitively costly to review these particular instances more carefully.

- Oracle: the test set is used as validation set. This answers the question "how would my detection method perform had I had access to an oracle that would give me perfect hyperparameters". Even if not useful in practical applications, this hyperparameter selection method gives us some indications on the behavior of detection methods.

**Baselines and normalization**  In order to properly evaluate the surveyed detection methods, we use the following baselines:

- None: No filtering step is performed, the training set is used as a whole including mislabeled examples.

- Random: Filtering or relabeling (depending on the experiment) uses random trust scores.

- Silver (perfect filtering): The training step only includes examples that have genuine labels.

- Gold: The whole training set is used, mislabeled examples are assigned their genuine label.

Since we compare detection methods across tasks of varying difficulties, we normalize by scaling the observed metrics (i.e. the test loss) linearly between 100 and 200 so that the performance of the none baseline gets 200 and the silver baseline gets 100.

**Machine learning models**  There are 2 different machine learning models involved in benchmarked pipelines: the base model used at the detection stage, and the final estimator of the pipeline (Figure 2). For both stages, we experiment with 2 different machine learning models: a kernelized linear model (KLM) trained with stochastic gradient descent and a gradient boosting model (GBT).

**Reproducibility**  Both detectors and helpers to download the datasets used in the benchmark are available in the open-sourced library described in section 4, available on the repository `https://github.com/Orange-OpenSource/mislabeled`.The benchmark code spanning from feature pre-processing to detector evaluation is available in a separate open-source repository `https://github.com/Orange-OpenSource/mislabeled-benchmark`, with fixed seeds for random number generators. The entire benchmark results are also available in a public archive `https://github.com/tfjgeorge/mis_bench_res`, so that reproducing figures and tables can be done without re-running the benchmark. We hope that all provided code and raw results will help foster research in weakly supervised learning.

## 5.2 Benchmark observations

This large scale benchmark allows us to ask a series of questions and observe some trends that we now highlight. For completeness, additional experiments with different setups (different final estimator), different noise structures (NCAR instead of NNAR), and different hyperparameter selection strategy (using a clean or noisy validation set) are deferred to appendix C.

**Overall performance of detection methods**  We start by evaluating detection methods in the detect + relabel pipeline (Figure 6). The experiment consists in relabeling the 10% less trusted examples as pointed by mislabeled detection methods. We use random trust scores as a baseline so that every setup is given the same number of training examples, and the same budget of relabeling. On most datasets, we observe an improvement in test loss compared to the random baseline, which confirms that mislabeled example detectors provide a useful signal. This also gives us a ranking between detectors on these particular tasks, where AGRA shows consistent performance compared to other methods.

**Overall performance of filtering pipelines**  We turn to detect + filter pipelines, and ask the question whether we can get an improvement in performance by using such a pipeline (which is fully automated and does not require additional human supervision) compared to just using a carefully regularized model on noisy dataset. This is not trivial as machine learning methods are known to already embed some form of robustness to noisy examples: by playing with hyperparameters that reduce its capacity, we can tune any machine learning method to focus on more salient features while trading off some flexibility to fit mislabeled examples. Furthermore, there is a trade-off between a filtered training set with only trusted examples, or keeping as many training examples as possible in order to have a bigger training dataset but at the cost of including potential mislabeled examples. In our experiments on NNAR, we observe that models trained on the subset of training examples that have genuine labels (the silver baseline) consistently get better generalization performance than models trained using the whole training set (the none baseline) including mislabeled examples (figure 7), even if there is less examples overall. More interestingly, we observe that using a pipeline detect + filter with hyperparameters tuned using a clean validation set allows to improve on generalization performance most of the time. For a few detectors, however, the detect + filter pipelines do not improve compared to the none baseline or the random baseline. We believe this could be improved by spending more effort to tune individual detectors.

**Detect/none capacity**  In figure 8, we compare the regularization hyperparameter given by the oracle in the classifier of the none baseline to the same hyperparameter in the classifier of detect + filter pipelines.

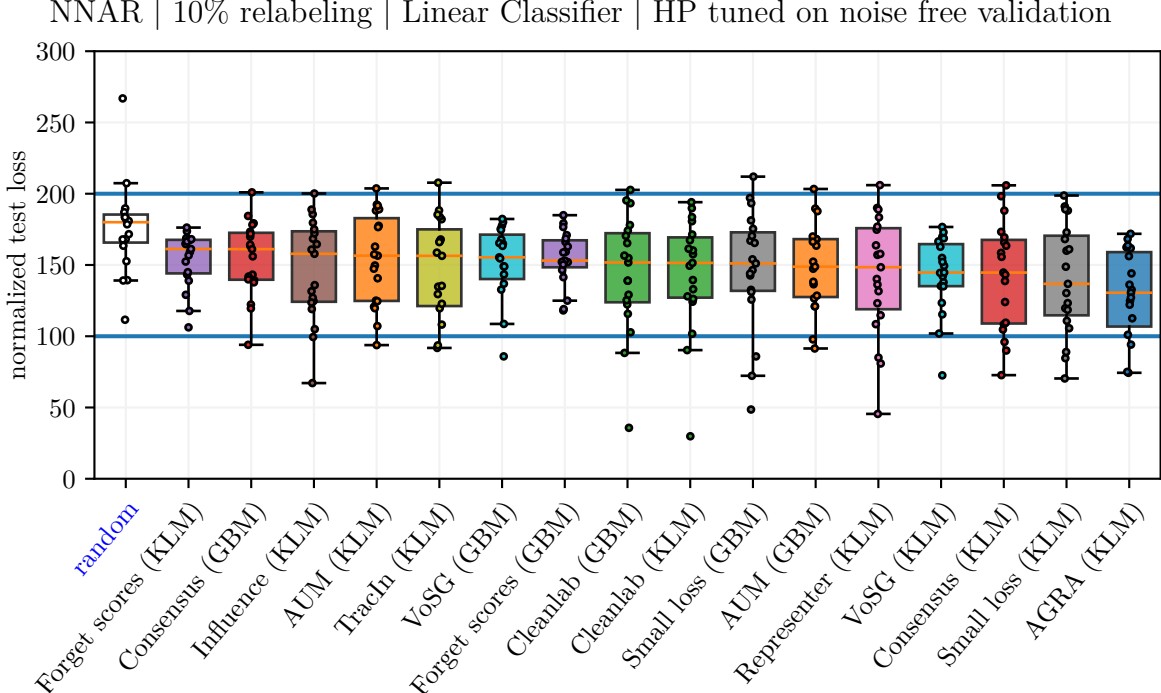

Figure 6: Distribution (boxplot) of the normalized (base 100=training on correctly labeled examples only, base 200=training on all examples including mislabeled ones) test loss of relabeling 10% less trusted examples with varying detectors using a linear model as estimator on tasks (dots) corrupted by NNAR. Hyperparameters are tuned using a clean validation set. Detector names include the detection method as well as the base model. Respective colors assigned to each detector are consistent across figures in the rest of this section.

We observe that most of the time, less regularization is needed in detect + filter pipelines, even if the trusted training set is smaller since untrusted examples have been removed. This is aligned with the intuition that noisy datasets require more robust machine learning models (i.e. with larger regularization).

**Clean or noisy validation set**   In our survey of mislabeled detection methods, we found that the question of the validation set was often overlooked: as for any machine learning application, the final performance crucially depends on a set of hyperparameter, among which the threshold used to filter untrusted examples is paramount. We found that choosing hyperparameters on a noisy validation set gave no improvement compared to the none baseline (figure 9). Intuitively, this can be understood as the fact that since the noisy training and noisy validation sets follow the same (noisy) distribution, from the perspective of the validation set, what is actually noise does not look like noise. In practice, the threshold for splitting the training set was often chosen to be 0 (no filtering at all, see figure 10). This questions the practical utility of mislabeled detection methods comparatively with the biquality setup (Nodet et al., 2021): in biquality learning, clean examples are used to actually learn the parameters of a model and simultaneously provide weak supervision on other (by default untrusted) examples whereas here, they are just used to choose hyperparameters, losing some useful signal.

**Detection performance vs final performance**   In previous experiments, we evaluated mislabeled detection methods by observing their performance in full pipelines (detect + filter or detect + relabel). This is the most relevant metric in practice since this is how detectors will be used in most cases. In this experiment, we instead evaluate detection methods by measuring the predictive power of trust scores to distinguish between genuine and mislabeled examples. We expect detectors that rank examples correctly would also lead to good classifiers trained on their most trusted examples. However, in figure 11, only a mild correlation is found between the two quantities, which suggests that good detectors possess other intrinsic qualities than their

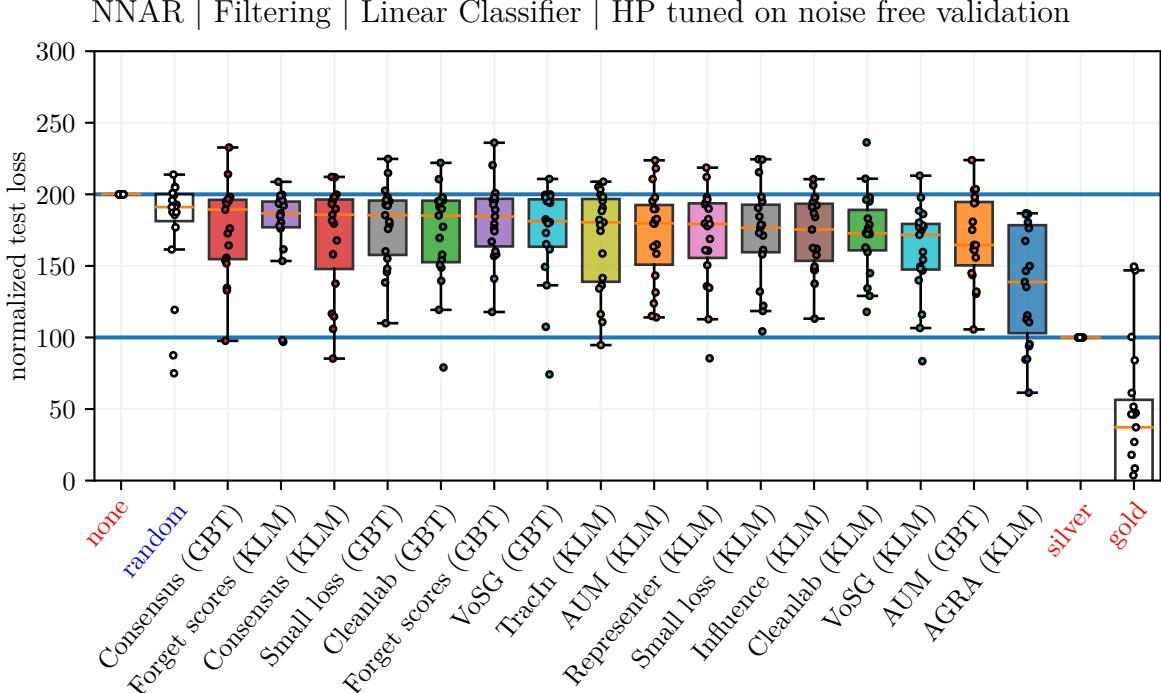

Figure 7: Distribution (boxplot) of the normalized (base 100=training on correctly labeled examples only, base 200=training on all examples including mislabeled ones) test loss of detect + filter pipelines with varying detectors with linear final estimator on tasks (dots) corrupted by NNAR. Hyperparameters are tuned using a clean validation set. Detector names include the detection method as well as the base model.

Figure 8: For each detector/dataset pair (a circle), we compare the oracle regularization ($\ell_2$ regularization in a linear model) chosen when using the whole corrupted training set on the x-axis, to the oracle regularization chosen in detect + filter pipeline on the y-axis. Most of the time, pipelines obtained a smaller regularization (dots are below the $y = x$ line).

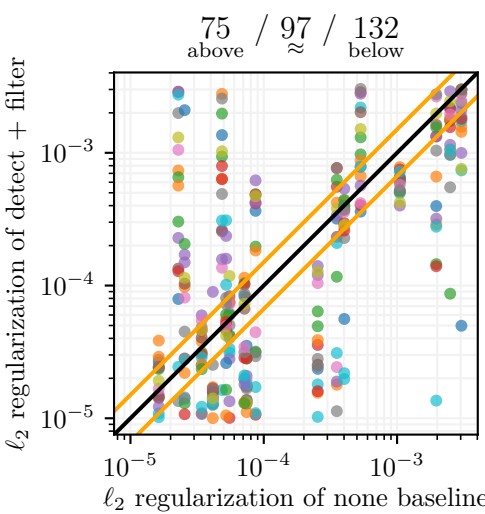

ranking capacity, such as their capacities to select prototype examples or balance datasets that are otherwise class-imbalanced.

**Weak is more difficult than noise**  We compare two sets of experiments on the same datasets: examples corrupted using artificial uniform noise (NCAR) and imperfect labeling rules (NNAR) in figure 12. Perhaps unsurprisingly, we observe that detect + filter pipelines perform worse on NNAR corruption than NCAR corruption. This is expected as NNAR is notoriously more difficult (Frénay & Verleysen, 2013): indeed, the patterns in the noise are often indistinguishable from the patterns in the non-corrupted data from the

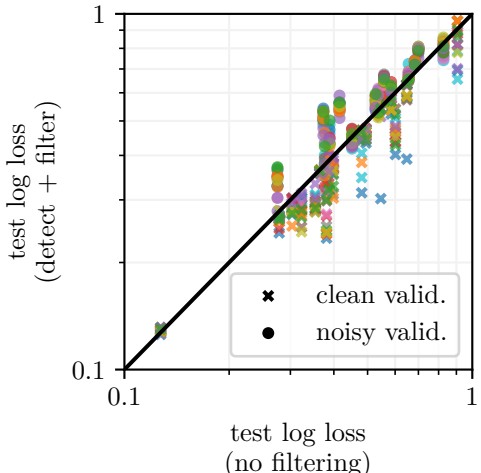

Figure 9: For each detector/dataset pair (a cross or a circle), we compare the classifier obtained by a detect + filter pipeline (y-axis) to the classifier trained on all examples, including noisy examples as a baseline (x-axis). Hyperparameters are tuned either on a clean validation set (crosses) or on a noisy validation set (circles) for both the baseline and the pipeline. We observe a trend where detect + filter pipelines tuned on the noisy validation set give worse performance than the baseline (circles are above the $y = x$ line), whereas pipelines tuned on the clean validation set give better performance than the baseline (crosses are below the $y = x$ line).

Figure 10: For each detector/dataset pair (a blue circle), we compare the split quantile obtained by cross-validation on a noisy validation set (y-axis) to the oracle (best) split quantile (x-axis). Choosing the split threshold using a noisy validation set consistently underestimates the optimal threshold. In fact, the 0 threshold (no filtering at all) is chosen most of the time.

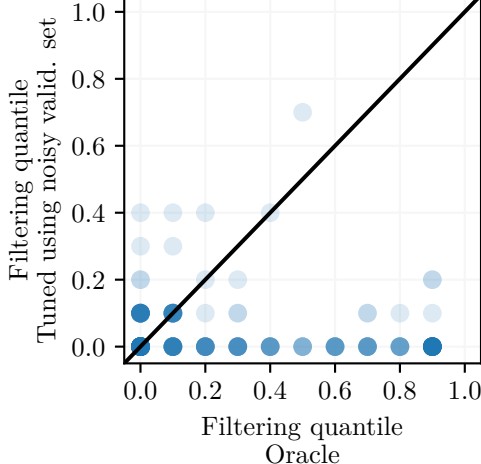

perspective of the learning algorithm in the absence of further hypotheses. More noteworthy, our experiments show no clear correlation between performance on NNAR and performance on NCAR. As a corollary, this questions the choice of testing mislabeled detection methods on NCAR-corrupted datasets: even if it is often more convenient to experiment on prevalent benchmarks and artificially introduce uniform label corruption, it might not translate to actual use cases with more intricate NNAR corruption.

**Adapting deep learning methods to classical machine learning algorithms** A contribution of this benchmark is also to evaluate mislabeled detection methods that were initially designed to work with deep learning models and that we adapted to work generically with any model that is learned sequentially (more details in appendix A). This is the case for AUM, Forget Scores, VoSG and TracIn. We evaluate them both with the gradient boosting algorithm and a linear model learned by stochastic gradient descent (SGD), which our library allows to treat as progressive learning natively. Empirically, we report mixed results: whereas some methods did not produce any significant improvement in test loss, VoSG with a linear model is among the best-performing methods. This demonstrates the feasibility of such an approach, which fosters future work on the similarities and differences of training dynamics between deep learning algorithms and other machine learning algorithms.

**Choice of base model and additive robustness** We hypothesize that in detect + filter pipelines, using different machine learning models as base model in the detection stage and as the final classifier could improve the final performance. The rationale is that different machine learning models include some form

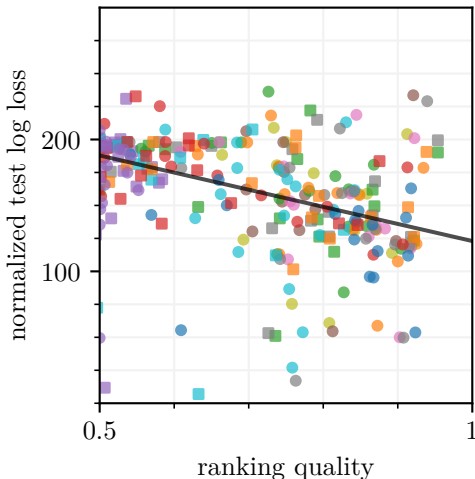

Figure 11: For each detector/dataset pair (a circle), we compare the normalized (none=200, silver=100) test loss of detect + filter pipelines on the x-axis, to the ranking quality of the trust scores on the y-axis. Detectors with better ranking tends to produce filtered dataset that allows the training of better classifiers (the black line is a robust linear regression).

Figure 12: For each detector/dataset pair (a circle), we compare the task with NNAR corrupted labels (x-axis) to the same training examples but corrupted using NCAR labels (y-axis). This confirms that NNAR tasks are more difficult than NCAR most of the time (circles are above the $y = x$ line). This also shows that there is no clear correlation between NCAR performance and NNAR performance.

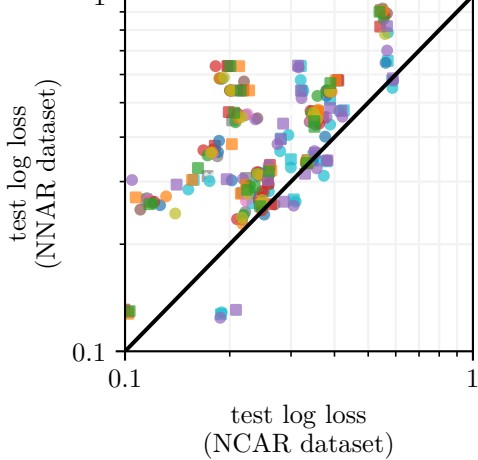

of robustness to different examples: we could benefit from the combined robustness by using 2 different methods. In figure 13, we compare the performance of detect + filter pipelines where both stages use either a linear model (KLM) or a gradient boosted tree (GBT) model. We expect to see two point clouds pulled away from the $y = x$ line, where KLM + GBT and GBT + KLM would perform better than KLM + KLM or GBT + GBT. We actually do not see such a pattern, which invalidates the hypothesis, at least in the current setup. Overall, it is clear from this figure that KLM is the best choice as a base model in these experiments.

**Representativeness of filtered data** Being able to accurately sort out mislabeled examples from a dataset is a fundamental property that detectors should possess, which they do (Figure 11). However, their ranking capacity does not explain by itself the performance of a model trained on a filtered dataset. We think that the representativity, in addition to the quality of the filtered dataset, matters. We experiment with a proxy of representativity, the class balance. We expect that the class balance of the filtered dataset to be closer to the test one than the noisy one. Figure 14 shows that most of the time, detectors tend to favor examples from the majority class, or in other words, filter out more aggressively examples from the minority class. We propose two potential reasons for this bias. Firstly, detectors may struggle to distinguish between mislabeled instances and those of the minority class, as the latter are inherently scarcer and more challenging to learn from. Secondly, the value among examples might not be evenly distributed. We speculate that failing to detect a mislabeled instance from the minority class could be more detrimental to the downstream taskthan failing to detect one from the majority class.

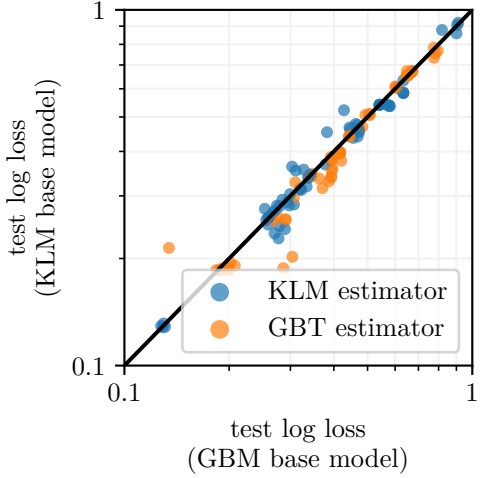

Figure 13: For each detector/dataset pair (a blue or orange circle), we compare (lower is better) the best classifier obtained when using a detector with a GBT model on the x-axis to the best classifier obtained when using a detector with a KLM model on the y-axis. KLM detectors (orange circles) seem to produce better performance most of the time (circles are below the $y = x$ line), and no clear pattern emerges as to whether mixing models show a trend (blue and orange circles do not show different patterns).

Figure 14: For each detector/dataset pair (a circle), we compare the class balance of the original training dataset on the x-axis to the class balance of the filtered train dataset on the y-axis. Orange circles correspond to datasets where training is less balanced than test, and blue circles correspond to datasets where training is more balanced than test. Even though the ratio of above and below circles varies by color, detectors tend to introduce more class imbalance than originally found in the training dataset (circles are below the $y = x$ line).

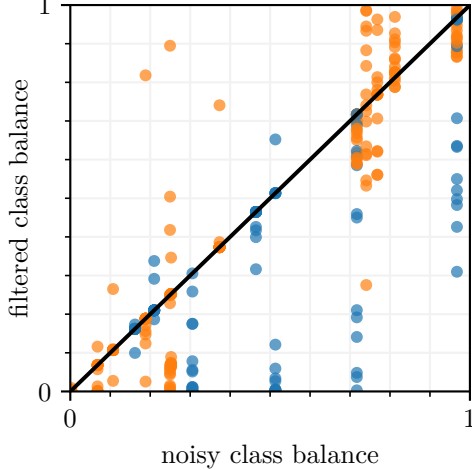

**Filtering by class** So far, we have studied detect + filter pipelines where the filtering step is performed regardless of the (potentially noisy) observed class of the example. As seen in the previous experiment, this can change the distribution of the class in the filtered dataset. Indeed, in class-imbalanced dataset, examples in the minority class often end up being less trusted than examples of the majority class. We thus experiment with the alternative strategy of filtering example class-by-class, where e.g. the top 50% most trusted examples of each class are kept for training. This ensures that the distribution between classes in the filtered dataset does not depart too much from that of the original (noisy) dataset, thus avoiding a situation where a minority class fully disappears from the training dataset. In Figure 5.2, we observe no such trend: for most tasks it does not make much difference, for some other tasks a tiny difference in performance is observed, either in favor of filtering by class, or in favor of filtering all classes at once.

**Detection performance vs base model performance** Similar to the fact that the inductive bias of machine learning algorithms is paramount to their classification performance on unobserved examples, we study how much this inductive bias relates to their performance at detecting mislabeled examples. Figure 16 shows that on datasets where the base model gets its better classification performance (measured as the test loss of the same model trained on the whole noisy training set), it is also better at detecting mislabeled examples. The trend is consistent across detectors.

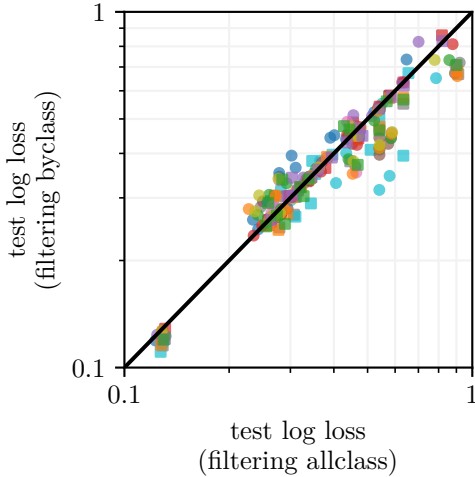

Figure 15: For each detector/dataset pair (a circle), we compare (lower is better) the best classifier obtained when filtering examples class-by-class on the x-axis to the best classifier obtained when filtering all classes at once on the y-axis. No clear picture emerges: it is sometimes better to filter class-by-class, sometimes better to filter all classes at once, and sometimes it does not make much difference.

Figure 16: For each detector across all datasets, we compare the performance at detecting mislabeled examples on the y-axis (AUROC for the task of detecting mislabeled examples, higher is better) to the performance of the underlying base model (measured using the test loss, lower is better) when trained using the whole training set including mislabeled examples. For each detector, we also plot a linear regression across datasets. We observe a trend where good classification performance correlates with good performance at detecting mislabeled examples

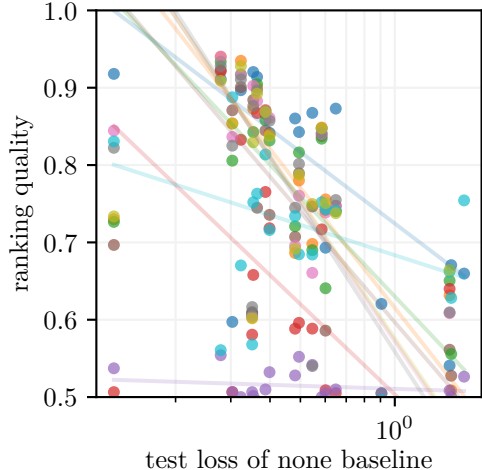

### 5.3 Lessons learned

Our benchmark offers a critical view on the approach of filtering in the context of machine learning with mislabeled examples when working with labeling functions or uniform noise: a fundamental flaw of such a methodology lies in the fact that it requires access to a clean validation set in order to select hyperparameters (the most important one being the threshold for splitting into trusted and untrusted examples). With this in mind, it seems like a waste of resources to use this clean data only for hyperparameter selection, rather than using it directly as a training set, or in a biquality learning setup. Filtering also imposes a hard threshold between trusted and untrusted examples, whereas in some cases, examples that are on the verge of the splitting threshold could also carry out some useful learning signal.

However, the results show some interesting trends that could inspire future research. In particular, it shows that some of the methods developed for deep learning algorithms (more specifically VoSG and AUM) also show promising results with other classical machine learning algorithms (here a linear model and a gradient boosting machine). On the contrary, Forget Scores do not seem to work very well with our current implementation. Since we could not afford to spend too much effort on every method, it could simply mean that we did not find appropriate hyperparameters, or that the training dynamics of deep learning on which Forget Scores rely are different from the training dynamics of GBTs and linear models.

Our experiments show encouraging results for applying model-probing methods to text and tabular datasets. For a small additional implementation cost, computing trust scores provides useful information about which

examples look genuine and which examples require additional reviewing. However, there is no clear winner among the detectors. Even though AGRA appears to perform better on this series of datasets, the distributions of normalized test log loss largely overlap, suggesting that the best performing detectors vary between datasets. This also provides room for improvement: since trust scores catch slightly different signals for different detectors, a natural extension might be to try and summarize several detectors into a single trust score.

## 6 Other related works

This paper is part of a series of surveys in the weakly supervised learning literature, specifically on learning with noisy labels (Frénay & Verleysen, 2013; Han et al., 2020; Song et al., 2022). However, it sets itself apart from other surveys by focusing on the task of identifying mislabeled examples instead of studying more broadly the literature of learning algorithms robust to noisy labels. It also provides a more modern view on the mislabeled example detection literature (Guan & Yuan, 2013) by proposing an encompassing framework closing the gap between approaches that were specifically developed for deep networks and classical machine learning algorithms.

Identifying mislabeled examples is a topic also found in the data cleaning literature (Ilyas & Chu, 2019) and has recently been successfully applied to the growing text datasets used to train or fine-tune large language models (e.g. Zhu et al., 2024). Data cleaning surveys (Côté et al., 2023) also have a broader scope than the one studied in this survey and are more comparable to the detect + filter pipeline but extended to other forms of data corruption such as feature noise, missing data, and outliers detection.

Furthermore, outlier detection is an important field related to mislabeled examples detection. These methods are designed to identify outliers in the sense $\mathbb{P}(X)$, whereas mislabeled detection methods seek to find outliers in the sense $\mathbb{P}(Y|X)$. Outlier detection approaches have been applied to split non-scalar trust scores, for example, when using the output of multiple detectors (Lu et al., 2023) or multiple probes where no apparent aggregation exists. Another use of outlier detection is to find outliers in $\mathbb{P}(X|Y)$ instead of $\mathbb{P}(Y|X)$ by training one outlier detection algorithm per class, assuming that outliers for a given class are mislabeled examples (Rebbapragada & Brodley, 2007). We did not include these methods in the survey, as they were out of scope.

Finally, we omit a series of methods that jointly optimize the two steps of the detect + handle pipeline. Most notably, these approaches work iteratively, akin to the expectation-maximization algorithm, where the detect and handle steps are optimized alternatively to minimize a global objective, usually getting the best possible classifier out of the handle step (Tanaka et al., 2018; Zeng et al., 2022). Contrary to the iterative refinement from section 3.3.1 where proper trust scores can be explicitly retrieved at every iteration, joint methods use implicit trust scores. As they only serve the role to guide the optimization procedure, they lack intrinsic significance, defeating our primary goal of mislabeled example detection.

## 7 Conclusion

The tremendous size of training datasets in modern tasks advocates for cheaper labeling strategies (e.g. crowdsourced annotation or automatic labeling rules), at the cost of some degree of labeling error. Methods for detecting mislabeled examples offer the promise of being able to diagnose training datasets and review examples in a second step, either automatically (using filtering, semi-supervised learning or biquality learning) or by manual relabeling. In this paper, we take a fresh look at past and recent detection methods, and show that most of them can be understood using a framework consisting of 4 components and a few principles. Notably this includes recent methods that exploit the particular training dynamics of deep networks, which we extended to be classifier-agnostic (in particular, in the empirical evaluation, we experimented with linear models and gradient boosted trees).

We proposed an implementation of this framework, that follows the scikit-learn API which is familiar to every machine learning practitioner. Our implementation focuses on the core mechanics of the framework, which then allows the implementation of the existing methods in the literature by only passing specific

values for the 4 components. This demonstrates that this framework is not just abstract but can also be actually implemented. Using this framework, we proposed a benchmark on a large number of tabular and text datasets, with some amount of labeling noise that comes either from uniform noise (NCAR noise) or from imperfect automatic labeling rules (NNAR noise). This benchmark is made available for reuse in the weakly supervised machine learning community, with helpers for automatically fetching datasets with train/validation/test splits and fixed seeds for pseudo-random generators.

This benchmark allowed us to provide a set of new insights for machine learning in the presence of mislabeled examples. We also empirically verify common folklore in the field, specifically the difference between NCAR and NNAR setups, or the fact that cleansing datasets from mislabeled examples allows to use less regularized models. We highlight the often overlooked issue of the role of a clean validation set free from any source of labeling noise in automatic pipelines, questioning their usefulness in real use cases: if I have access to clean examples, why not use them directly to learn the parameters of my model using a semisupervised algorithm or biquality learning?

### Perspectives

The framework presented in section 3, and the implementation that we distribute, suggest that there is room to improve the detection of mislabeled examples by experimenting with combinations of base model, probe, ensemble strategy, and aggregation methods that have not yet been explored. We encourage a systematic comparison of probes by looking specifically at which examples they score differently. Depending on the context, it may be interesting to use a mix of probes in order to improve the robustness of detection methods. More generally, the idea of using trained machine learning models in order to diagnose datasets of examples extends to other subfields of machine learning, such as active learning, example-based explainability methods, or conformal prediction. This urges for cross-fertilization between communities for a toolbox of methods that extend machine learning algorithms in order to obtain not only a single prediction (e.g. class or a real value) but also additional information about that prediction.

Finally, we advocate for a more fine-grained score of each instance than just being correctly or mislabeled. As we discussed briefly in section 2.4, training examples can be categorized depending on whether they belong to a region of the input space with low or high aleatoric or epistemic uncertainty, and recent works aim at capturing this distinction (e.g. Javanmardi et al. (2024) using conformal prediction). Moreover, some examples might be more useful or harmful, either because they belong to a minority class or an underrepresented pattern in an otherwise class-balanced dataset, or because they ward against some undesired bias of the training data. In some cases, these are the examples that are the most important ones, for instance, in fairness sensitive tasks. A more relevant metric would then be to use a measure of how useful is any given example to the prediction of a learned classifier on test examples of this minority group, such as the DataShapley value (Ghorbani & Zou, 2019) using a custom utility function.

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

# A Implementation details and comments regarding specific detection methods

## A.1 Variance of gradients

We found a small inconsistency in the experiments in Agarwal et al. (2022): the toy experiment involves a MLP with no non-linearities, which computes a linear mapping of the input vector. In this case, the gradient of the logit w.r.t. the input vector only depends on the equivalent weight matrix $W = W_1 W_2 \ldots W_l$, and not on the input. Put differently, the proposed VoG statistics is the same for every example, which means that it cannot be used to rank examples. We thus think that there is an inconsistency in the results presented in the toy experiment section. This is in contrast to the larger scale experiment with ResNet architectures, where the mapping from the input space to the logits is non-linear since it involves non-linearities (here ReLU activations).

When working with linear models in our experiments, we instead implemented a slightly different version where we differentiate the probability given by the softmax, and not the logit. This mapping from input space to softmax output is non-linear, and it depends on the example contrarily to the mapping from input space to logit. We found the resulting statistics (i.e. the variance of gradients of the probability given by the softmax) to be useful at detecting mislabeled examples.

## A.2 Finite differences

During our survey, we reviewed some detectors which where fundamentally tailored to work on differentiable machine learning models. For example, the Variance Of Gradients detector probes the model by looking at the derivative of the pre-softmax layer of a neural network with respect to the input features. We proposed in the library to use the finite difference approach for non-differentiable models such as decision trees. On the same 2D toy-dataset used in the original paper (Agarwal et al., 2022), the finite difference method showed to reasonably approximate the exact method for a kernelized linear model:

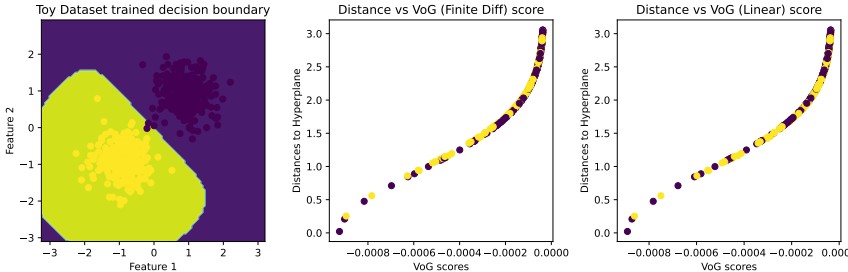

Figure 17: We reproduce the experiments from figure 1 in Agarwal et al. (2022) with a kernelized linear model and finite difference approximation.

Thus it can be instantiated with the user's favorite progressive ensemble such as gradient boosted trees.

# B Benchmark details

## B.1 Generating noisy label from labeling rules

The datasets used in the benchmark provides, on top of ground truth labels for each examples, a set of weak labels given by labeling rules. Given an example, a labeling rule output a weak label, that may be a class label if the rule matched, or nothing. These weak labels are aggregated thanks to a majority vote. However two edge cases may arise, the first if the votes are tied between two classes, the second when no labeling rules matched for an example. In the first case we pick a class completely at random among tied winners, for the latter we chose to drop these examples from the training dataset.

Dropping unmatched examples is not a consensus among the weakly supervised learning literature, the usual approach is to assign them a random label. We think dropping is a more sensible approach, akin to what practitioners would do in practice, than the random assignment.

It should be noted that such noisy labels generate a non-squared transition matrix as the set of noisy classes is equal to the set of original classes plus the unlabeled class. In the figures from table 2, the unlabeled noisy class corresponds to the last row of the transition matrix $\mathbf{T}$.

For example, on the *youtube* dataset, the noise transition matrix is the following:

$$\begin{pmatrix} 0.79 & 0.22 \\ 0.04 & 0.74 \\ 0.17 & 0.04 \end{pmatrix}$$

The columns corresponds to true labels and the rows corresponds to noisy labels. The element in the second row and first column means that 4% of examples from class 0 have been assigned the noisy labels 1. The last element of the last row means that 4% of examples from class 1 have been assigned no noisy labels.

The implementation of weak label encoding is available in the open-source library, see section 4.

## B.2   Feature engineering

In order to have a fair starting ground between the different machine learning models used in section 5, all datasets have been preemptively encoded to only contain numerical attributes.

Two different feature engineering pipelines are applied to each dataset, depending if it's a text or tabular dataset.

For text datasets, TF-IDF features (Schütze et al., 2008) are generated and $\ell_2$ normalized. The size of vocabulary is chosen on a per dataset basis to balance between accuracy and compute time. For tabular datasets, categorical features are one-hot encoded and numerical features are normalized.

Each component of the feature engineering pipeline uses scikit-learn's implementation.

## B.3   Details regarding machine learning models

Two families of models are used through the benchmark, kernelized linear models (KLM) and gradient boosted trees (GBT), two of the most popular approaches for machine learning on tabular data.

To have scalable KLMs, we chose to use the Random Kitchen Sinks approach (Rahimi & Recht, 2007) to approximate kernel computations on large-scale datasets. We used the Gaussian RBF kernel (Broomhead & Lowe, 1988) for tabular datasets and linear (or no) kernel for text datasets. Then, the linear model is trained by minimizing the log-loss on training samples thanks to Stochastic Gradient Descent. All KLMs components use scikit-learn's implementation.

For GBTs, we chose the CatBoost (Prokhorenkova et al., 2018) implementation mainly because of its fast training time on GPUs.

## B.4   Hyperparameters Sampling

We summarize the search space used in random search of the main hyperparameters of each family of models described in section B.3 in the following table:

On top of that models from both families are trained for a maximum of a thousand iteration (number of epochs for KLM, number of trees for GBT), ensuring convergence in most cases. Yet, models can be early stopped if their log-loss on an holdout dataset does not decrease for more than 5 iterations. This specific holdout dataset corresponds to 10% of the training data.

|  | Hyperparameter | Search space |
|---|---|---|
| KLM | kernel bandwith | $\{\frac{1}{\phi(d)\mathbb{V}(\Phi(x))}\}$ |
|  | $\ell_2$ regularization | log-uniform $[1e-5, 1e-1]$ |
|  | learning rate | log-uniform $[1e-3, 1]$ |
| GBT | $\ell_2$ regularization | uniform $[0, 100]$ |
|  | learning rate | log-uniform $[1e-5, 1e-1]$ |

Table 3: Table of hyperparameters.

## C   Additional results

### C.1   Relabeling

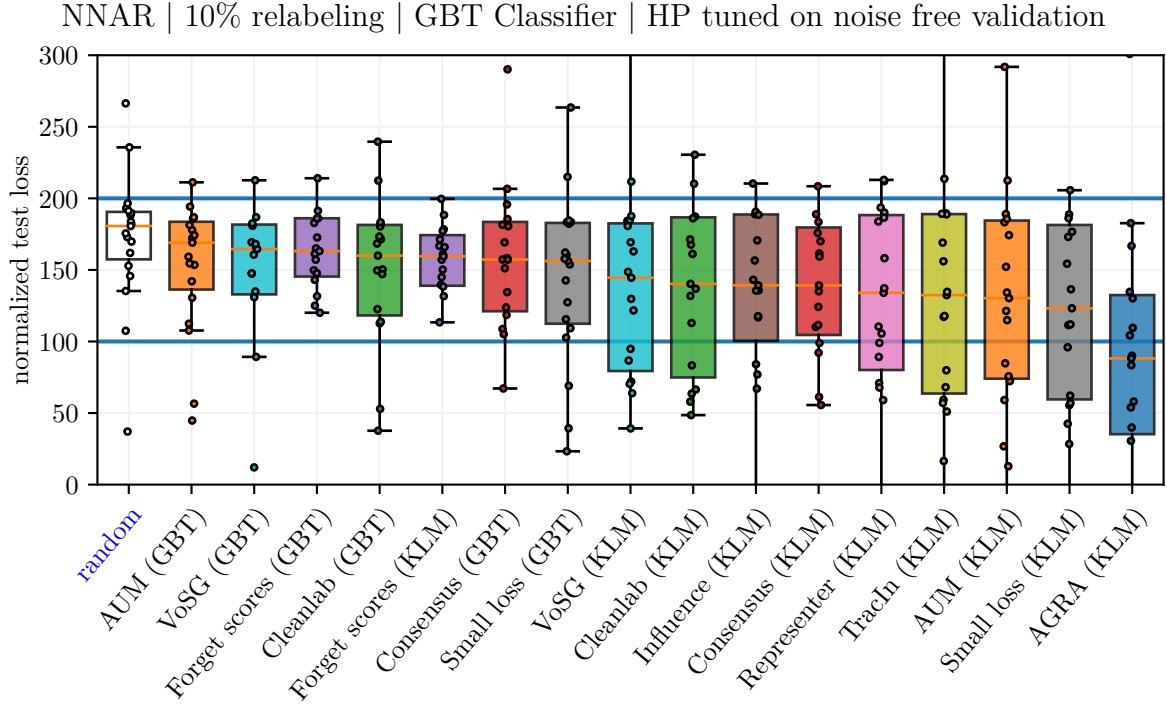

Figure 18: Distribution (boxplot) of the normalized (base 100=training on correctly labeled examples only, base 200=training on all examples including mislabeled ones) test loss of relabeling 10% less trusted examples with varying detectors using a GBT model as estimator on tasks (dots) corrupted by NNAR. Hyperparameters are tuned using a clean validation set. Same as figure 6 but with a GBT estimator.

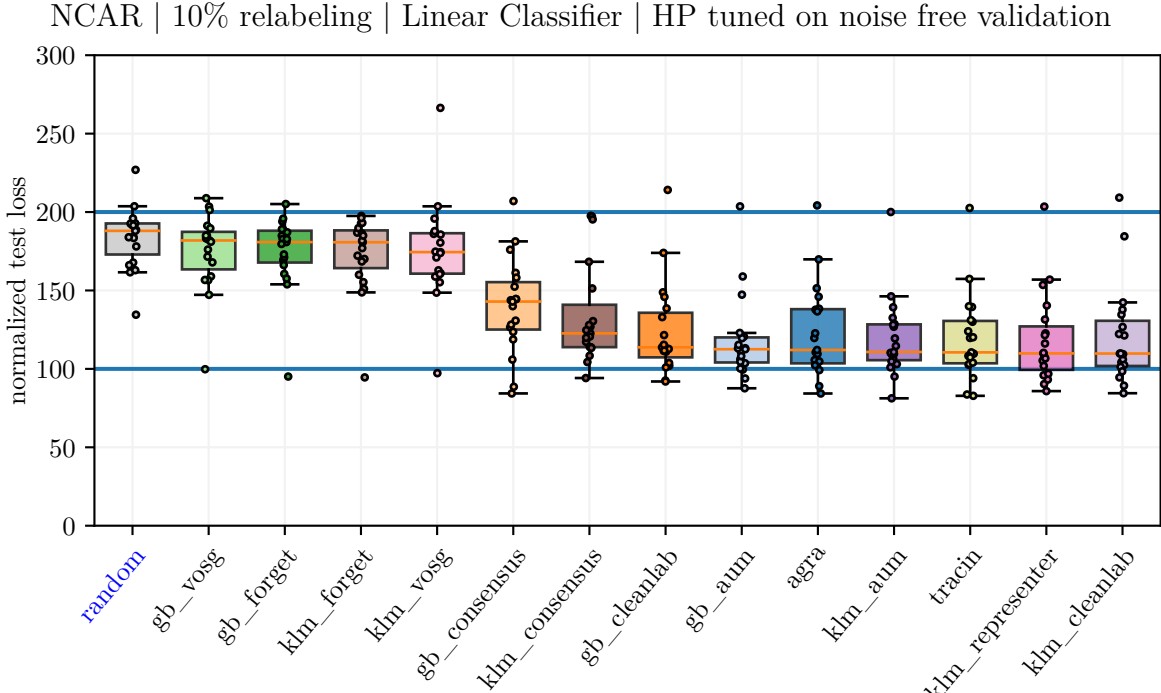

Figure 19: Distribution (boxplot) of the normalized (base 100=training on correctly labeled examples only, base 200=training on all examples including mislabeled ones) test loss of relabeling 10% less trusted examples with varying detectors using a linear model as estimator on tasks (dots) corrupted by NCAR. Hyperparameters are tuned using a clean validation set. Same as figure 6 but with a NCAR noise.

## C.2 All detectors

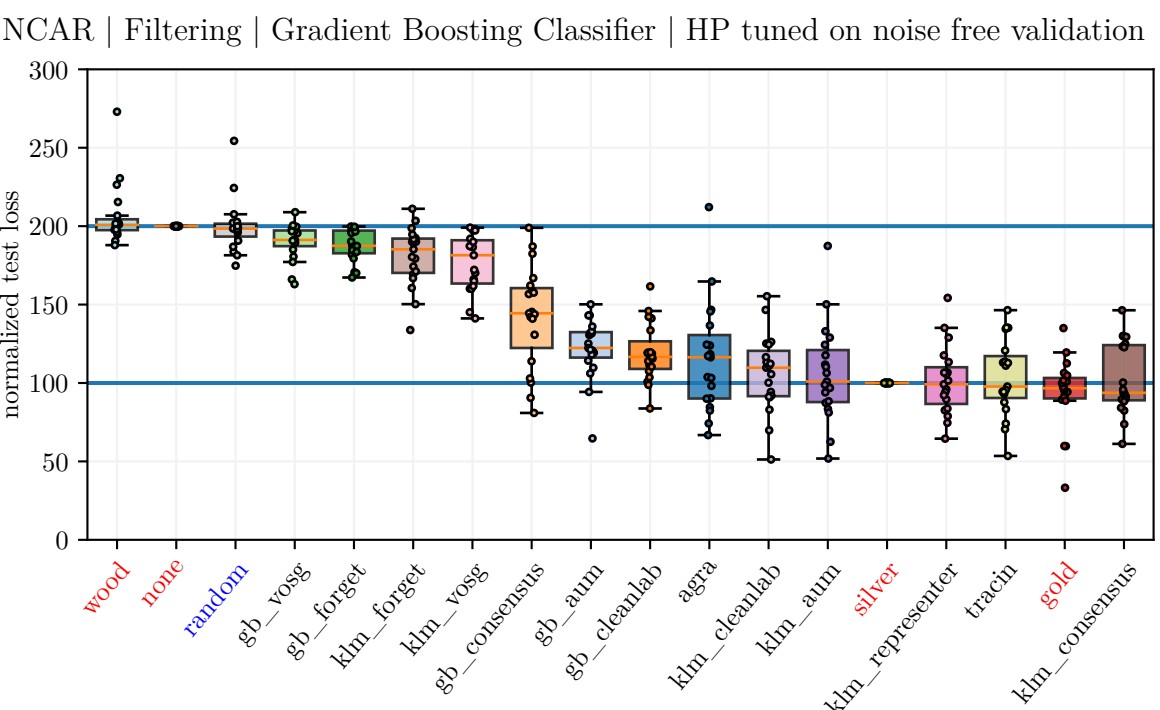

Figure 20: Normalized test loss of detect + filter pipelines with varying detectors with GBT final estimator on tasks corrupted by NCAR. Hyperparameters are tuned using a clean validation set. (same as figure 7 but using a GBT final classifier)

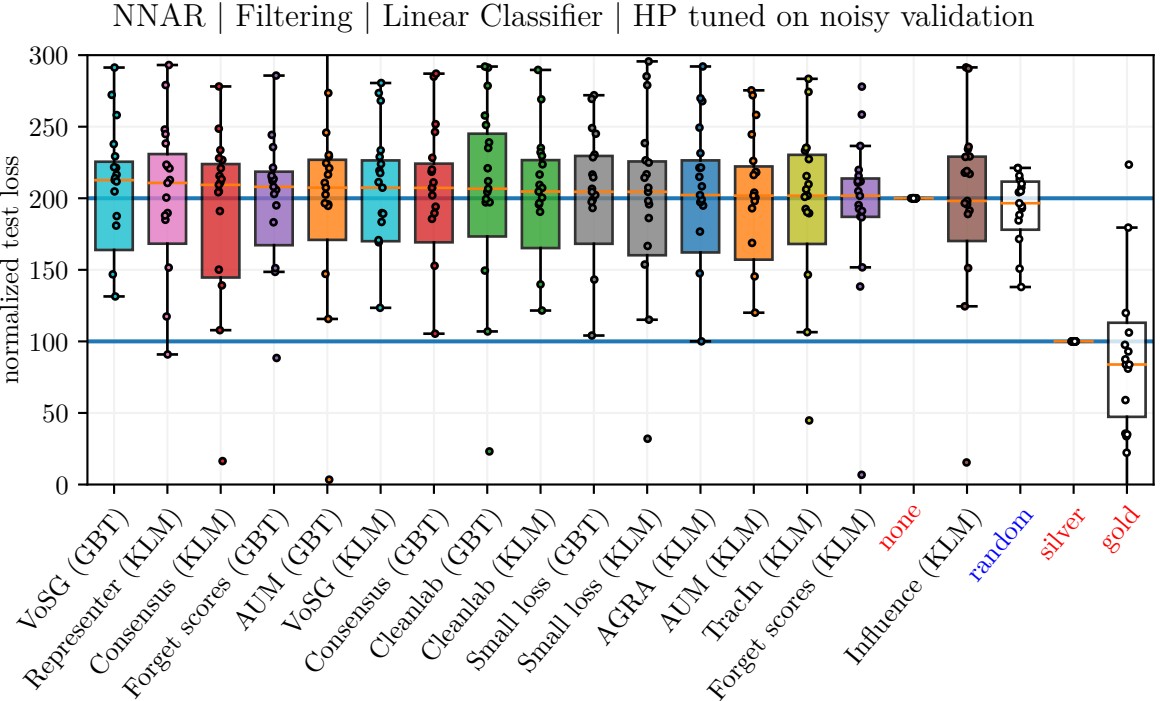

Figure 21: Normalized test loss of detect + filter pipelines with varying detectors with linear final estimator on tasks corrupted by NNAR. Hyperparameters are tuned using a noisy validation set. (same as figure 7 but using a noisy validation set for hyperparameter selection)

### C.3 Filtering threshold - noisy validation set vs oracle

Figure 22: For each detector/dataset pair (a blue circle), we compare the split quantile obtained by cross-validation on a noisy validation set (y-axis) to the oracle (best) split quantile (x-axis). Choosing the split threshold using a noisy validation set consistently underestimates the optimal threshold. In fact, the 0 threshold (no filtering at all) is chosen most of the time. (same as figure 10 but using a GBT final estimator)

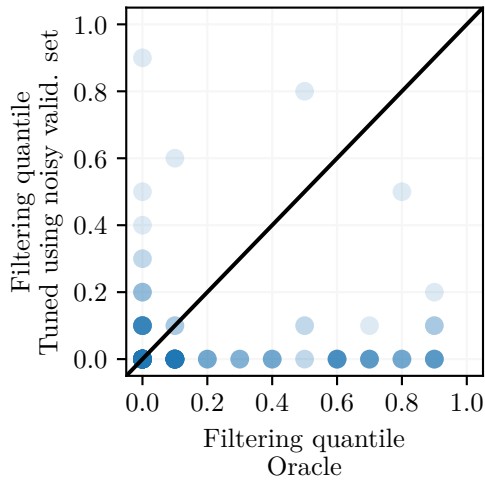

### C.4 Regularization of none vs pipelines

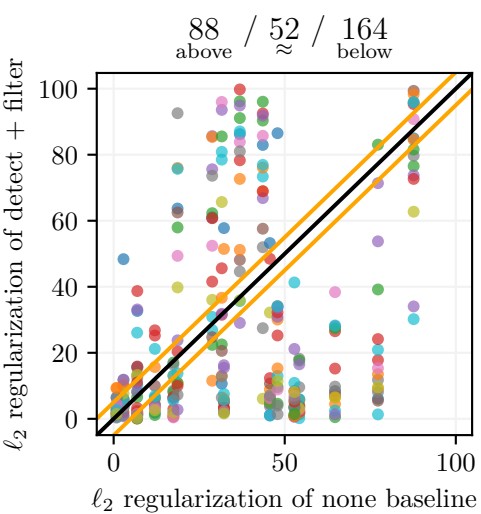

Figure 23: For each detector/dataset pair, we compare the oracle regularization ($\ell_2$ regularization in a GBT model) chosen in detect + filter pipeline, to the oracle regularization chosen when using the whole corrupted training set. Most of the time, detect + filter pipelines obtained a smaller regularization, meaning that filtering noisy examples allows for less regularized classifiers. (same as figure 8 using a different final estimator)

# D Other data modalities

While we mainly focus on text and tabular data as these represent an important use case of machine learning techniques, we include the following additional experiments, where we showcase our library applied to different data types.

## D.1 Results on a regression task on tabular data

We experiment on the California Housing regression task, that consists in predicting the price in the housing market (a real number) using several features of the sold house. Here the same framework of section 3 can be used provided that we use a regression probe. We use a model-probing detector with a random forest regressor as base model, bootstrapping as the ensemble strategy, the $\ell_2$ loss as the probe and the mean across out-of-bag examples as the aggregation method. As illustrated by the bottom-5 trusted examples circled in red in figure D.1, the detector correctly identifies some examples with suspicious labels.

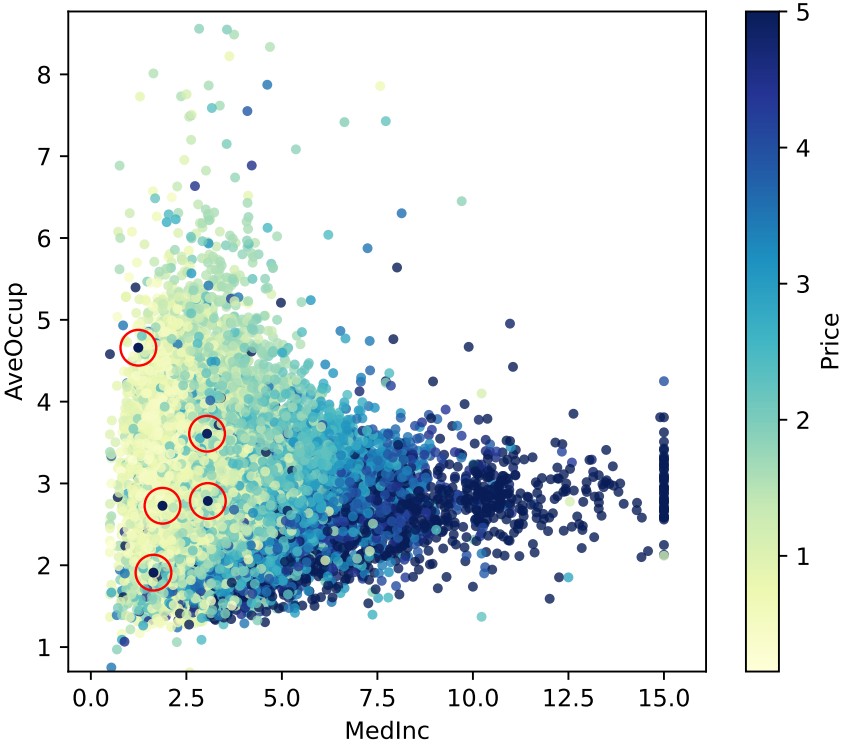

Figure 24: Bottom-5 trust scores on the California housing dataset

## D.2 Results on an image classification task with features from a pre-trained ResNet

We use the 50.000 images of the CIFAR10 classification task. We extract features before the classification layer of a ResNet50 model, pre-trained on the ImageNet dataset. Using these features, we choose a LogisticRegression model from scikit-learn with default hyperparameters as our base model, and TracIn as our detector.

In figure 25, for each class, we show the less trusted image (in red), as well as 4 other images of the same class. As expected, the most untrusted image often looks less representative of the observed class, or even mislabeled (e.g. a windsurfer in class "ship", a dog that looks like a cat, or a zoomed-in image of the front of a truck whereas other images of trucks often display full trucks with their trailer).

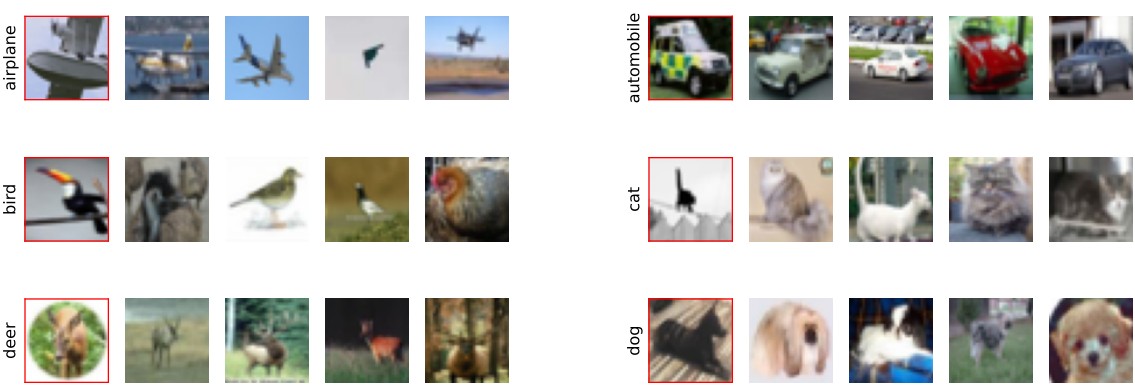

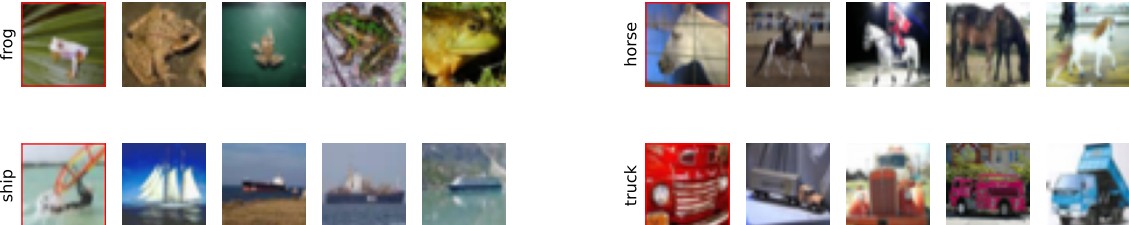

Figure 25: On CIFAR10, for each class we display the less trusted training set image (red frame), compared to 4 other representative images from the same class.

## D.3 Results on an image classification task with confusing classes

We use 10.000 images from the Animal-10N classification task (Song et al., 2019). The setup is similar to the previous section D.2, where we extract features before the classification layer of a ResNet50 model, pre-trained on the ImageNet dataset. Using these features, we choose a LogisticRegression model from scikit-learn with default hyperparameters as our base model, and TracIn as our detector.

In figure 26, for each class, we show the less trusted image (in red), as well as 4 other images of the same class. Each row represents very similar classes (e.g. cats often look like lynxes, wolves like coyotes, etc). In this setup, less trusted images correspond to very unusual pictures, such as drawings of a cheetah and a wolf. In contrast, other images are real photographs, or a coyote seating in a bus whereas other coyote images have more usual backgrounds.

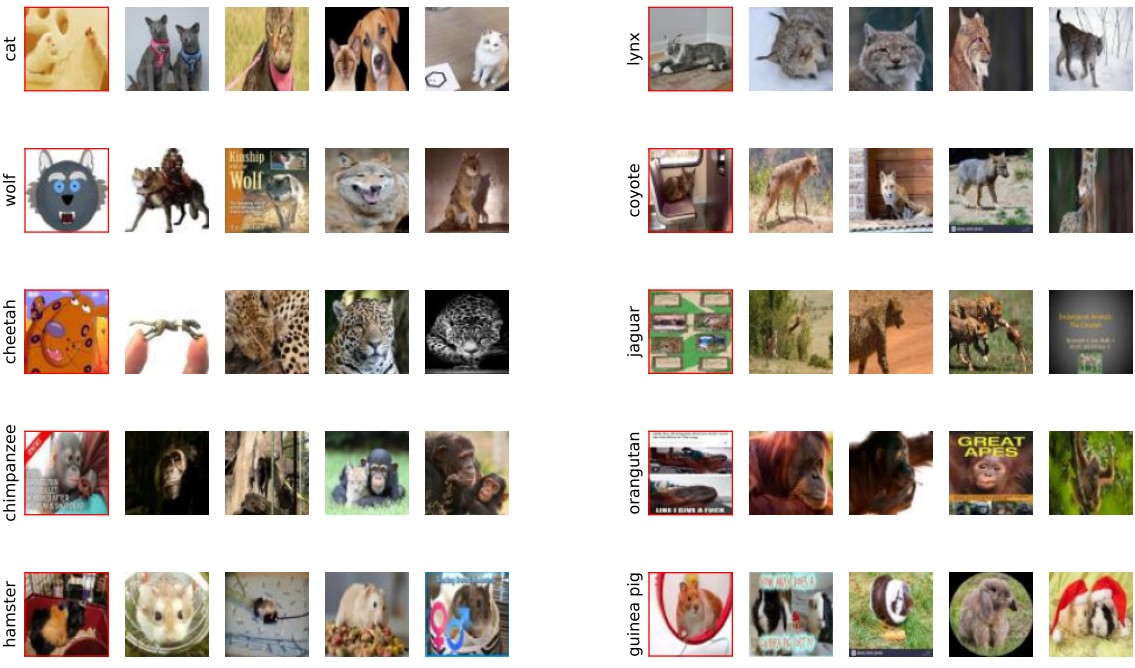

Figure 26: On Animal-10N, for each class we display the less trusted training set image (red frame), compared to 4 other representative images from the same class. Left and right classes correspond to similar-looking classes that are more likely to be confused.

