# OpenReview forum: "Mislabeled examples detection viewed as probing machine learning models: concepts, survey and extensive benchmark"
_TMLR — Accepted by TMLR_

### Review · Reviewer_AptK · 2024-08-08

**Summary Of Contributions:**

1. Comprehensive survey for mislabeled examples detection in text and tabular data, including model, model probe, ensemble strategy, and aggregation method.
2. Providing a python library that integrates mislabeled examples detection framework, making the whole framework clear.
3. Providing a benchmark on text and tabular data, and conducting comprehensive experiments and analysis on the benchmark.

**Audience:**

Yes

**Claims And Evidence:**

No

**Requested Changes:**

1. Expand the discussion on mislabeled examples in other modality data, especially vision data, including semi-supervised visual methods and their relation to fully supervised methods. Also, there are some interesting tasks, such as concealed object detection, that may provide more insights. This would make the survey more comprehensive and informative.
2. More discussion about mislabeled examples' influences on different tasks. For example, the use of unfiltered train data in large language models and multimodal large language models, and the potential issues arising from data generated by large language models and multimodal large language models. This will provide a more in-depth understanding of the challenges and opportunities of mislabeled example detection for various tasks. For example, halluciation in large language models and multimodal large language models now seriously affects models' performance.

**Strengths And Weaknesses:**

Strengths:
1. Clear theoretical derivation and formula presentation.
2. A large number of experiments and visualizations of charts for clear visualization of methods and results.
3. Providing a Python library, contributing to the community, and future work.

Weaknesses:
1. Limited discussion on vision data: The article mainly focuses on tabular and text data, with limited discussion on other modalities data, for example, vision data.
2. Insufficient discussion of mislabeled example detection applications. For example, data used to train large language models and data generated by large language models which are current research hotspots.

---

> ### Author Response · Authors · 2024-08-23
>
> We would like to thank the reviewer for their time and consideration at reading our paper.
>
> We here answer their comments, and we updated the PDF accordingly.
>
> ## Other data modalities and LLMs
>
> We would like to emphasize that our purpose was precisely to focus on tabular and text data as this is, we think, explicitly stated in the abstract and the introduction of the paper. Indeed, these modalities are less popular in today’s main ML conferences, despite being frequently used in industry (see e.g. a recent position paper [1]). We consider vision and LLM applications to be outside of the scope of the current paper. Additionally, focusing on text and tabular data allowed us to use classical ML algorithms (specifically, kernelized linear models and gradient boosted trees) with fixed features, rather than relying on a deep model ‘s ability to learn a good representation.
>
> However, to answer your requested changes, we added an experiment using CIFAR10 in appendix D.2, with features extracted from a pretrained ResNet model, as an illustration than the same principles hold with this type of data.
>
> Regarding LLM training data, we already mentioned it as a potential use case of mislabeled detection methods in section 2.5. We updated the paper with an extended discussion of the importance of mislabeled examples in LLM training data in section 2.5 “Use cases”
>
> ## Claims and evidence
>
> We noticed that you ticked “No” in the Claims and Evidence section of your review. We think that is because a great part of the paper is mostly a survey. But if this is not the case: Could you please be a bit more specific regarding which claim in the paper is not supported by evidence in your view?
>
> [1] Van Breugel and Van Der Schaar, “Position: Why Tabular Foundation Models Should Be a Research Priority.”, ICML 2024

---

> ### Author Response · Authors · 2024-09-11
>
> Dear Reviewer,
>
> Can you please point out the claims that you feel are not supported by evidence since the last revision of our paper?
>
> The deadline for your final recommendation is approaching, and we would appreciate the opportunity to clarify any remaining issues.

---

### Review · Reviewer_rwvM · 2024-08-19

**Summary Of Contributions:**

1. The paper introduces a novel modular framework for detecting mislabeled examples in datasets. This framework is parameterized by four core components, providing a unified perspective that encompasses most existing mislabeled detection methods.

2. A comprehensive survey of existing mislabeled detection methods is provided, where these methods are categorized and compared within the proposed framework. The survey includes methods designed for both deep learning models and classical machine learning algorithms.

3. The authors contribute a Python library that implements the modular framework.

4. The paper includes an extensive empirical evaluation, benchmarking a variety of mislabeled detection methods on datasets with different types of labeling noise.

**Audience:**

Yes

**Claims And Evidence:**

Yes

**Requested Changes:**

Please revise the paper according to the comments above.

**Strengths And Weaknesses:**

**Strengths**

1. The paper presents a unified, modular framework for detecting mislabeled examples in datasets, and releases a library for fast implementation and benchmarking.

2. The paper offers a thorough review of existing mislabeled detection methods, providing valuable insights into their strengths and weaknesses.

3. The empirical results reveal important findings, such as the limitations of existing methods in handling complex noise structures and the potential benefits of certain detection strategies. These insights are valuable for guiding future research in this area.

**Weaknesses**

1. Dependency on Clean Validation Sets: The effectiveness of the detection methods often hinges on the availability of clean validation sets for hyperparameter tuning. In real-world scenarios where such clean data may not be accessible, the practical utility of these methods could be significantly constrained.

2. Omission of Recent Works: The paper's survey, as illustrated in Table 1, primarily focuses on methods developed no later than 2023, with most benchmarked detection methods dating before 2022. To offer a truly comprehensive survey, it is essential to include more recent works, such as the detection method for text data discussed in [R1]. Furthermore, given the paper's emphasis on text data, benchmarking "natural" label noise in popular RLHF datasets, as examined in [R2], would be particularly relevant.

3. Information Loss in Stochastic Case Handling: In Section 2.1.2, titled "Stochastic Case: The True Concept is Defined as a Probability Distribution," the paper discusses using a threshold to convert a probability into a hard label. This approach results in significant information loss, potentially limiting the effectiveness of the method.

[R1] Unmasking and improving data credibility: A study with datasets for training harmless language models. ICLR 2024.

[R2] Training a helpful and harmless assistant with reinforcement learning from human feedback. 2022.

---

> ### Author Response · Authors · 2024-08-23
>
> We first thank the reviewer for their time at reading and reviewing our paper.
>
> We here answer their comments, and we updated the PDF accordingly.
>
> ## Dependency on Clean Validation Sets
>
>  We 100% agree with the reviewer regarding the constrained practical utility of such methods in certain scenarios. In fact, this is a message that we want our paper to convey, as we think this is an issue that is often overlooked in previous literature. This is explicitly mentioned in the first paragraph of section “5.3 Lessons learned”.
>
> Despite this limitation, mislabeled detection methods are still useful in other contexts such as active relabeling, corresponding to our first experiment (Figure 6), or integrated in a more complete pipeline such as your reference [R1].
>
>  ## Omission of Recent Works
>
> Thank you for pointing to this recent work. With regards to R1, its detection stage (which is the focus of our paper) re-uses the method described in Zhu et al., 2022 that we already referenced.  We added a reference in section 6 as we recognize it is a promising approach.
>
> With regards to your other reference R2, we agree that this is another interesting use case, especially given LLMs’ current popularity. We, however, think it is out of scope for the current paper which focuses on tabular and text data with classical ML models.
>
> ## Information Loss in Stochastic Case Handling
>
> We agree that this definition in the stochastic case may result in information loss. The purpose of this section is also to highlight the difficulty in giving a formal definition in this case, and we are aware of the shortcomings of this definition that we explicitly state in the last paragraph in section 2.1.2. However, we note that the threshold tau can be chosen arbitrarily small, in which case only classes with 0 probability are considered mislabeled. We updated this last paragraph to more clearly reflect this potential information loss.

---

> > ### Comment · Reviewer_rwvM · 2024-09-05
> > **Thanks for the rebuttal**
> >
> > Thanks for the rebuttal. The information loss still exists even if the tau is small. If tau means a probability, any threshold will break the meaning of probability. Although using a threshold is a common approach, I wonder if the author has any other ideas to handle the probability with minimal loss of information.

---

> ### Author Response · Authors · 2024-09-09
>
> Thanks for engaging in the discussion.
>
> By choosing tau=0, we get no loss of information, since only 0 probability classes under the true distribution are considered mislabeled. Depending on the task this could be a sensible choice. The downside is that if we only have a few samples, and unfortunately they are from the minority class e.g. in a 10%/90% mixture under the true distribution, then this could mislead the ML algorithm into predicting the minority class with high confidence.
>
> We tried to look at this question from all angles, and we came up with the definition in the paper. We also attempted at explicitly stating the limitations of this definition.

---

> > ### Comment · Reviewer_rwvM · 2024-09-26
> >
> > Thanks for the explanation. I guess there is a misunderstanding between us. If an event happens with some probability, we should not simply treat it as a deterministic event. In other words, simply treating them as a deterministic event incurs information loss. It can be a future direction.

---

### Review · Reviewer_tFQi · 2024-08-21

**Summary Of Contributions:**

This paper considers the problem of detecting mislabeled data and proposes a general framework composed of four modular components that can integrate various existing techniques. The main contribution is a Python library that precisely implements these principles and benchmarks existing methods against both random and non-random label noise across a series of datasets with automated labeling rules. The experiments are conducted fairly to me, and the key findings offer valuable insights for advancing research in this area.

**Audience:**

Yes

**Claims And Evidence:**

Yes

**Requested Changes:**

It is recommended to test the proposed framework on real-world datasets with noisy labels, such as the Animal10N dataset, to strengthen the applicability of this work in practical scenarios.

**Strengths And Weaknesses:**

Strengths:
1. This paper proposes a general framework for noisy label detection that can integrate various existing techniques. The Python implementation provides valuable support for research in this field.
2. The empirical studies are thorough, usding various benchmark datasets, and the findings are convincing.

Weaknesses:
1. The proposed framework follows a basic procedure for noisy label detection. While the empirical findings are valuable, the theoretical contribution is somewhat limited. Additionally, the ensemble module is primarily an engineering solution rather than a novel theoretical advancement.
2. The experiments are primarily conducted on benchmark datasets with artificially generated label noise. The performance of the proposed method with different modules remains unclear on real-world noisy labeled datasets, such as the Animal10N dataset [1].

[1] Song, H., Kim, M., and Lee, J., "SELFIE: Refurbishing Unclean Samples for Robust Deep Learning," In Proc. 36th Int'l Conf. on Machine Learning (ICML), Long Beach, California, June 2019

---

> ### Author Response · Authors · 2024-08-23
>
> We would like to warmly thank the reviewer for their time and suggestions.
>
> We here answer to their comments, and updated the PDF accordingly.
>
> ## Artificially generated label noise
>
> We would like to clarify that most experiments shown in the main document actually use real label noise, which comes from imperfect labeling rules, a frequent use case in industry [2]. While we agree that it does not comprehensively cover all types of labeling noise (which would be a gigantic effort), it is different from artificially introduced uniform or class dependent random noise.
>
> The additional experiments on uniform (NCAR) noise, deferred to the appendix, are only here for completeness, and Figure 12 in the main text for shows the different performance at mislabeled detection methods between artificially introduced noise, and actual noise. We updated the abstract to better reflect the difference between the NCAR and NNAR experiments and highlight that NNAR experiments come from actual noisy datasets.
>
> ## Other data modalities
>
> Including an experiment with an image dataset was also suggested by reviewer AptK. We reproduce our answer to them here:
>
> > We would like to emphasize that our purpose was precisely to focus on tabular and text data as this is, we think, explicitly stated in the abstract and the introduction of the paper. Indeed, these modalities are less popular in today’s main ML conferences, despite being frequently used in industry (see e.g. a recent position paper [1]). We consider vision and LLM applications to be outside of the scope of the current paper. Additionally, focusing on text and tabular data allowed us to use classical ML algorithms (specifically, kernelized linear models and gradient boosted trees) with fixed features, rather than relying on a deep model ‘s ability to learn a good representation.
>
> However, we followed your suggestion, and we added a proof-of-concept experiment on the dataset Animal-10N which we report in section D.3 in appendix. The highlighted untrusted examples here are unusual examples, such as drawings whereas most other examples are real photographs.
>
> [1] Van Breugel and Van Der Schaar, “Position: Why Tabular Foundation Models Should Be a Research Priority.”, ICML 2024
> [2] Ratner, A., De Sa, C., Wu, S., Selsam, D., and Ré, C., “Data Programming: Creating Large Training Sets, Quickly”, NeurIPS

---

> > ### Comment · Reviewer_tFQi · 2024-09-15
> >
> > Thanks for the response. The revised draft has addressed my main concerns.

---

### Author Response · Authors · 2024-08-23
**PDF update**

We thank the reviewers for their constructive feedback.

We updated the paper accordingly, with the following changes:
 - New proof-of-concept experiments on CIFAR10 and Animal10N image datasets
 - Extended discussion regarding LLM data use-cases
 - Addition of references to recent work
 - Correction of typos

---

### Decision · Action_Editor_GMkJ · 2024-09-28

**Recommendation:** Accept as is

**Comment:**

All three reviewers were unanimously in favour of acceptance. Following the discussion, one remaining comment was that the paper is rather long; the authors may wish to consider whether some material can be relegated to the Appendix, to aid the reader.

**Audience:**

Mislabeled example detection is of fundamental importance in problems involving label noise, and reliable machine learning. The survey in the paper is of clear interest to those working in these areas. The new framework and library is also expected to be of interest, with the latter having the potential to facilitate systematic evaluations of new methods proposed for the problem.

**Claims And Evidence:**

The paper's claims are that it provides a detailed survey and new view of several methods to detect mislabeled examples, as well as a new library for assessing methods on a range of benchmarks. These claims are supported by detailed discussion and analysis.